# CLM5-FruitTree: a new sub-model for deciduous fruit trees in the Community Land Model (CLM5)

**Olga Dombrowski**[1], **Cosimo Brogi**[1], **Harrie-Jan Hendricks Franssen**[1], **Damiano Zanotelli**[2], **and Heye Bogena**[1]

[1]Agrosphere (IBG-3), Forschungszentrum Jülich GmbH, Jülich 52425, Germany
[2]Faculty of Science and Technology, Free University of Bolzano, Bolzano, 39100, Italy

**Correspondence:** Olga Dombrowski (o.dombrowski@fz-juelich.de)

**Abstract.** TS1 The inclusion of perennial, woody crops in land surface models (LSMs) is crucial for addressing their role in carbon (C) sequestration, food production, and water requirements under climate change. To help quantify the biogeochemical and biogeophysical processes associated with these agroecosystems, we developed and tested a new sub-model, CLM5-FruitTree, for deciduous fruit orchards within the framework of the Community Land Model version 5 (CLM5). The model development included (1) a new perennial crop phenology description, (2) an adapted C and nitrogen allocation scheme, considering both storage and photosynthetic growth of annual and perennial plant organs, (3) typical management practices associated with fruit orchards, and (4) the parameterization of an apple plant functional type. CLM5-FruitTree was tested using extensive field measurements from an apple orchard in South Tyrol, Italy. Growth and partitioning of biomass to the individual plant components were well represented by CLM5-FruitTree, and average yield was predicted within 2.3 % of the observed values despite low simulated inter-annual variability compared to observations. The simulated seasonal course of C, energy, and water fluxes was in good agreement with the eddy covariance (EC) measurements owing to the accurate representation of the prolonged growing season and typical leaf area development of the orchard. We found that gross primary production, net radiation, and latent heat flux were highly correlated ($r > 0.94$) with EC measurements and showed little bias ($< \pm 5\%$). Simulated respiration components, sensible heat, and soil heat flux were less consistent with observations. This was attributed to simplifications in the orchard structure and to the presence of additional management practices that are not yet represented in CLM5-FruitTree. Finally,

the results suggested that the representation of microbial and autotrophic respiration and energy partitioning in complex, discontinuous canopies in CLM5 requires further attention. The new CLM5-FruitTree sub-model improved the representation of agricultural systems in CLM5 and can be used to study land surface processes in fruit orchards at the local, regional, or larger scale.

## 1 Introduction

Orchards and other perennial fruit crops are a major component of the global agricultural production, with significant coverage and yield in China, the United States, south-western Africa, and some parts of Europe (FAO, 2021). In the European region, perennial crops are a key economic element of Mediterranean agroecosystems as they provide 45 % of the local agricultural output (Lobianco and Roberto, 2006). Apples are the most important fruit tree crop, as one-third of European orchards is devoted to their production. With a coverage of 984 509 ha, they provide a yearly harvest of over 17 million tons, which is one-fifth of the overall European fruit production in terms of output value (FAO, 2021).

In contrast to annual crops, fruit trees can be productive for several decades before rotation is needed. Their prolonged growing season, standing biomass, and low respiratory losses can support carbon (C) storage and promote higher C use efficiencies (Wünsche and Lakso, 2000; Zanotelli et al., 2013). The transport of C stored in biomass into the soil and reduced soil tillage and disturbances under fruit orchards compared to annual crops further promote C sequestration (Bwalya, 2012; Wu et al., 2012; Ledo et al., 2020). The FAO has therefore

suggested perennial agriculture as a possible measure to mitigate climate change and enhance food security (Glover et al., 2010), and many studies have recently investigated this potential for various fruit orchards (Wu et al., 2012; Scandellari et al., 2016; Hammad et al., 2020; Yasin et al., 2021). The study of water and irrigation requirements in fruit orchards has become another field of intense research due to the need for a more resilient agriculture in the context of climate change and water supply shortages (Maestre-Valero et al., 2017; El Jaouhari et al., 2018; O'Connell and Scalisi, 2019; Segovia-Cardozo et al., 2022). In order to answer questions related to C sequestration, water requirements, and sustainable food production of fruit orchards, a better understanding of the related ecosystem processes is vital (Fader et al., 2015).

Models with a comprehensive description of the carbon, water, and energy fluxes, such as global land surface models (LSMs), are a powerful tool to explore complex ecosystems like the abovementioned fruit orchards. The use of LSMs was recently extended to not only model the processes at the land–atmosphere interface, but also to study the response of ecosystems and water resources to climate change (Prentice et al., 2015; Fisher and Koven, 2020; Blyth et al., 2021). To quantify these effects, LSMs need to represent a wide range of land use and vegetation types. However, most LSMs consider only perennials such as deciduous and coniferous trees as well as major annual crops such as wheat, soy, or maize (Lawrence et al., 2018). Recently, some LSMs additionally included bioenergy crops (Schaphoff et al., 2018), while others group crops into a few generic crop types (Noilhan and Mahfouf, 1996; Krinner et al., 2005; Balsamo et al., 2009). Despite their significance, perennial crops, such as fruit trees, are rarely considered in LSMs, and attempts to include them in global and regional modelling environments are scarce (Fader et al., 2015; Cheng et al., 2020). An example of such an attempt is the inclusion of agricultural trees (e.g. grapes, cotton, and apple trees) in the Lund–Potsdam–Jena managed Land (LPJmL) model to improve the representation of Mediterranean agroecosystems (Fader et al., 2015). Here, agricultural trees were modelled as small trees, and fruit harvest was determined as the product of a plant-specific harvest index and the net primary productivity (NPP). Other authors parameterized oil palm trees, a perennial evergreen crop, in the Community Land Model (CLM) version 4.5 (Fan et al., 2015). Palm trees were represented by a new phenology where large palm leaves with fruit bunches emerge successively, leaves are pruned regularly, and harvest occurs once a month. Recently, two perennial grasses for energy production were parameterized in the latest version of the model, CLM5 (Cheng et al., 2020). Parameters for bioenergy crops were tuned using sensitivity analysis and observations, while harvest was represented by removing around 70 % of the aboveground biomass.

While the abovementioned studies describe some common features of perennial plants, they do not, or only partially, represent the seasonal deciduous phenology of fruit trees or the explicit modelling of fruit growth. Furthermore, key aspects such as C reserve accumulation and mobilization in the following spring are generally not considered, possibly due to necessary simplifications or because the drivers of these processes are still not fully understood (Le Roux et al., 2001; Neumann, 2020). The absence of perennial crops in LSMs introduces a significant bias in the representation of biogeophysical and biogeochemical processes in agroecosystems where this type of cultivation is prevalent. As a result, the response to climate change in terms of C sequestration, water requirements, or food production cannot be assessed adequately in regions such as the Mediterranean, where perennial, woody crops are very common and play a vital role in food security and economy (Fader et al., 2015; Lobianco and Roberto, 2006).

Although deciduous fruit trees share certain characteristics with natural vegetation and annual crops in LSMs such as CLM5, several particularities in their growth dynamics and management practices still prevent a meaningful simulation using currently available representations of vegetation. In this study, we therefore provide CLM5 with the ability to model perennial fruit trees and the associated processes. For this purpose, we developed a new sub-model named CLM5-FruitTree within the existing model framework of CLM5. CLM5-FruitTree combines elements of the broadleaf deciduous tree subroutine such as growth and C turnover of woody components, with distinctive phenological stages and a harvestable organ similar to the annual crop subroutine. We first describe the model conceptualization including the new phenology, carbon and nitrogen (CN) allocation, and management options. We further demonstrate the applicability of CLM5-FruitTree by parameterizing a new apple plant functional type (PFT). Finally, we evaluate and discuss the model performance using extensive field data from an apple orchard in South Tyrol, Italy.

## 2 Methods

### 2.1 Vegetation characterizations in CLM5

The latest version of the Community Land Model, CLM5, simulates the exchange of water, energy, C, and nitrogen (N) between land and atmosphere as well as their storage and transport on the land surface and in the subsurface, driven by climate variability and modulated by soil and vegetation states and characteristics. The land surface in CLM5 is characterized by one of five land units, namely glacier, lake, urban, vegetated, and crop. These units are further divided to capture the variability in soil, vegetation, and management options (i.e. irrigated or non-irrigated). Compared to previous model versions, CLM5 features various improvements in the representation of land use and vegetation modelling,

such as plant CN cycling, soil and plant hydrology, and crop modelling (Lawrence et al., 2018; Lombardozzi et al., 2020).

Many of the C and N cycle components of CLM5 were originally derived from the Biome BioGeochemical Cycles (Biome-BGC) model (Thornton et al., 2002). Here, vegetation is represented conceptually by three different plant C and N pools that are maintained separately for the individual plant organs (leaf, live/dead stem, fine root, live/dead coarse root, and grain). The storage pools represent C and N reserves, the transfer pools serve as intermediate pools to separate fluxes in and out of the storage pools, and the display pools represent the actual growth of a given organ (Fig. 1). C made available through photosynthesis is first used to support maintenance respiration of live organs based on organ N content, temperature, and a constant base rate as proposed by Atkin et al. (2015). Dead stem and dead coarse root components are assumed to consist of dead xylem cells, without metabolic function (no C cost for maintenance). The remaining C can then be allocated to the growth of new tissue considering associated growth respiration costs. Maintenance respiration, growth respiration and C cost of N uptake from the soil comprise the autotrophic respiration component ($R_a$) in CLM5. Plant material reaching the end of its lifespan feeds into different litter pools from where it progressively decomposes to soil organic matter under C losses through heterotrophic respiration ($R_h$).

For the simulation of fruit orchards, a module for perennial deciduous crops is needed, which is currently missing in CLM5. Such a module must account for the perennial deciduous nature of fruit trees, which is similar to the existing representation of broadleaf deciduous trees (BDTs) included in Biome-BGC but with differences in phenological triggers, vegetation structure, and C partitioning. In addition, it must represent growth and harvest of the fruits and typical management practices, of which some are already conceptualized in the prognostic Biogeochemistry Crop Module (BGC-crop), while others are not yet implemented. The algorithm for the seasonal phenology of BDT controls initial leaf development and senescence that mark the beginning and end of a growing season based on temperature and day length thresholds. Once a new growth period is initiated, C and corresponding N fluxes accumulated in the previous season occur out of the storage pools into the transfer pools, from where they are gradually sent to the display pools (Fig. 1). During the active growth period, C and corresponding N storage pools are replenished based on specified C : N ratios of each plant organ. During leaf senescence, C and N pools feed the litter or coarse woody debris pool except for live stem and live coarse roots that are mostly retained as structural woody tissue (dead stem and dead coarse roots).

BGC-crop, adopted from the prognostic crop module of the Agro-Ecosystem Integrated Biosphere Simulator (Agro-IBIS), currently features eight different annual crop species with interactive crop management options (i.e. irrigation and fertilization). Another 23 currently inactive crop types can be defined but have not been provided with specific crop parameters (Lombardozzi et al., 2020). Crop phenology and CN allocation follow three phenological phases: (1) from planting to leaf emergence, (2) from leaf emergence to the start of grain fill, and (3) from grain fill to grain maturity and harvest, which are controlled by temperature and growing degree-day (GDD) thresholds. Different to natural vegetation, crops have a grain pool representing the harvestable organ but no structural woody tissue. Furthermore, all assimilates are directed to the displayed pools, while the storage pools remain unused. At harvest, C and N from the grain pool are transferred to a grain product pool, while a small amount is kept to reseed the crop in the following year. All remaining plant parts feed the litter cycle (Fig. 1). The reader is referred to Lombardozzi et al. (2020) and the technical documentation of CLM5 for a more detailed description of the BDT and crop representation (Lawrence et al., 2018).

From the above description of the existing vegetation modules, the following limitations for the application of CLM5 to deciduous fruit trees arise. (1) The current BGC-crop algorithm does not allow the simulation of perennial and/or woody crops. (2) The BDT phenology algorithm, although describing some characteristics common to fruit trees, lacks the capability to simulate a harvestable organ, individual development of different plant parts, and the separation of growth from C reserves of the previous year and photosynthetic growth of the current season. (3) Typical management practices of fruit orchards such as transplanting of tree seedlings and pruning are currently not represented in CLM5. (4) There is no parameterized fruit tree PFT in the default parameter set of CLM5.

## 2.2 Model conceptualization and technical implementation

To resolve the model limitations discussed in Sect. 2.1, we developed a new sub-model, CLM5-FruitTree, to model the ecosystem processes and exchanges of energy and matter of deciduous fruit trees grown in commercial orchards, with a focus on the simulation of biomass growth and yield. More specifically, for the implementation of CLM5-FruitTree, we introduced a new phenology subroutine that describes the main phenological development of fruit trees and includes triggers for seasonal orchard management practices typical under organic or conventional production. In addition, the CN allocation module as well as corresponding modules (C and N state and flux updates) were modified to reproduce the growth dynamics of fruit trees and to model the fates of C and N in the orchard system. The sub-model development does not include any changes to the existing calculation schemes for radiative transfer or momentum, heat, and water fluxes to explicitly account for the discontinuous canopy structure of tree rows and vegetated or non-vegetated alleys in fruit orchards. In-row and between-row planting distances and alley vegetation are not defined directly. Instead, the orchard struc-

ture and the area covered by the canopy are accounted for through parameterization of the leaf and stem area indices, the planting density, maximum canopy height, and aerodynamic parameters, similar to the implementation of crops and forest in CLM5.

CLM5-FruitTree combines characteristics of both BDT and annual crops to simulate a perennial woody crop with a harvestable organ making use of the existing concepts of storage, transfer, and display vegetation pools described in Sect. 2.1 (Fig. 1). Similar to the existing BDT phenology algorithm in CLM5, the fruit tree algorithm uses a perennial deciduous phenology with standing woody biomass and annual leaf shedding. During the active growth period, however, the phenology and CN allocation of vegetative and harvestable organs are described by distinct growth phases and are driven by a GDD summation similar to the crop phenology.

An orchard is established by transplanting small tree seedlings from a nursery, a typical planting method for this type of cultivation (Wheaton et al., 1990; Corelli-Grappadelli and Marini, 2008). Once planted, the orchard remains productive according to a user-defined lifespan which, depending on fruit tree type and production system, typically ranges between 10 and 30 years (Demestihas et al., 2017; Cerutti et al., 2014). The sub-model makes no specific assumptions about the rootstock, but the effect of different rootstocks in terms of tree height and rooting depth can be set by the user via the respective parameters, *ztopmx* and *root_dmx* (Table C1). In CLM5-FruitTree, both stored C and current photosynthesis contribute to the growth of the fruit tree, as leaf and shoot development at the beginning of a growing season utilizes carbohydrate reserves and nitrogenous compounds that were accumulated during the previous season (Tromp, 1983; Oliveira and Priestley, 1988; Loescher et al., 1990). Deciduous fruit trees are dormant in winter and resume growth in spring after meeting species- and cultivar-specific chilling and heat requirements (Anderson et al., 1985; Faust et al., 1997; Zavalloni et al., 2006), which is represented in CLM5-FruitTree using the chilling and forcing model proposed by Cesaraccio et al. (2004). Early in the season, the canopy develops rapidly until it reaches maturity typically by midsummer, while leaf shedding occurs when temperatures drop in autumn (Kozlowski, 1992; Loescher et al., 1990; Lakso et al., 1999). Fruit trees usually start flowering 3–4 weeks after bud break, which is not specifically represented by CLM5-FruitTree, which instead assumes that fruit growth begins at the end of flowering (Lakso et al., 1999). The implementation of flowering to include effects of non-optimal pollination, frost during flowering, or hormonal processes affecting fruit set and development is outside of the scope of this development and of minor importance for large-scale simulations and processes at ecosystem level that are typically the focus of LSMs such as CLM5. Consequently, CLM5-FruitTree does not produce information on fruit size or number but only on total yield, which we consider ade-

quate for most applications of the sub-model development. Fruit growth is described by two stages, cell division and cell expansion that together form a sigmoid growth curve observed for many fruit tree species such as apple, pear, and orange (Corelli-Grappadelli and Lakso, 2002; Jackson, 2011).

In the following, the new developments to account for the distinct phenology, CN allocation, and management practices of a fruit orchard are described in more detail. Other biochemical and biophysical processes such as photosynthesis, water and litter cycles, and fixation and uptake of N were not modified except for minor adaptations to the re-translocation of N and respiration to enable the use of certain parts of these scripts for the fruit tree PFT. The technical implementation of some features of the new phenology routine (transplanting, pruning, harvest, and final rotation) was based on CLM-Palm, a previous model development for palm trees in CLM4.5 (Fan et al., 2015, and unpublished code). References where code elements were directly reused or modified based on CLM-Palm are made in the published source code of CLM5-FruitTree (Dombrowski, 2022). Along with the new sub-model, an apple PFT was parameterized using one of the existing but thus far inactive crop types in CLM5, types 35 and 36 (rainfed and irrigated citrus).

### 2.2.1 Phenology

A new orchard life cycle is initialized by transplanting seedlings at the beginning of the year during dormancy. Tree growth thereafter is described by six post-planting phenological stages, namely (1) bud break, (2) fruit growth, (3) fruit ripening, (4) canopy maturity, (5) fruit maturity and harvest, and (6) start of leaf senescence (Fig. 2).

Bud break is predicted by a sequential model that first accumulates chill days followed by anti-chill days based on a predefined temperature threshold and chilling requirement (Cesaraccio et al., 2004). More information on the sequential model and the calibration of model parameters can be found in Appendix A. Outside the dormant period, leaf and fruit development occurs in parallel but with a time shift as fruit growth typically starts 4–5 weeks after bud break, while canopy development continues until mid-season and leaf senescence does not occur until after the fruits are harvested (Wünsche and Lakso, 2000; Goldschmidt and Lakso, 2005) (Fig. 2).

The thermal thresholds to reach phases (2)–(5) are defined as accumulated GDDs since bud break and can be adjusted by the user via the parameter file, which applies to all parameters listed in Table C1 of the Appendix. GDDs are determined as the difference between the average daily air temperature and a base temperature of 4 °C with a maximum daily increment of 26 degree days (Eq. 1). Different to the existing deciduous phenology, leaf senescence is triggered not by day length but by the drop of the daily mean temperature below a critical temperature threshold, in this case the base temperature. This approach was selected since many fruit trees that

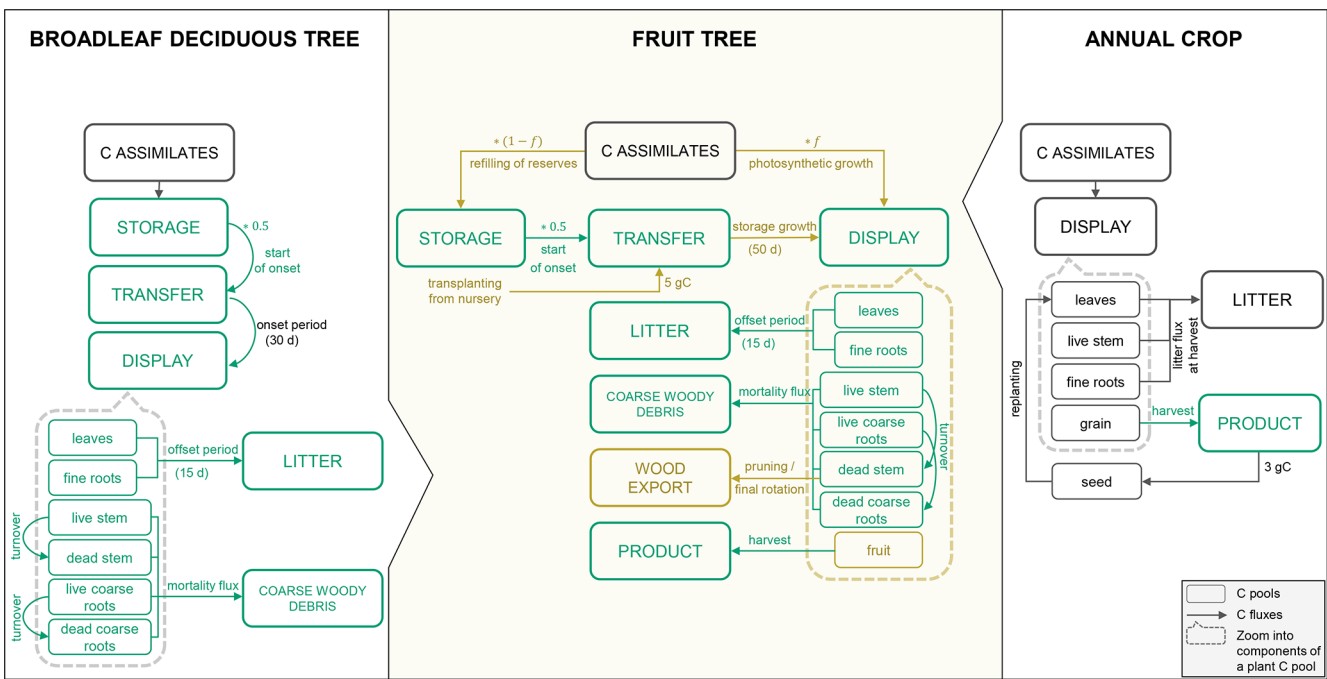

**Figure 1.** Schematic of the main phenology and C allocation features of the broadleaf deciduous tree and annual crop representations in CLM5 as well as the new CLM5-FruitTree sub-model. C pools within the dashed boxes are the individual components that make up the displayed C pool (the same components can be found for the other main plant pools: storage and transfer pools, respectively). Carbon pools and fluxes in green were reused for CLM5-FruitTree, while pools and fluxes in brown were modified or newly added.

belong to the Rosaceae family (e.g. apple, pear, plum, and cherry) are unaffected by photoperiod and are instead controlled by temperature (Heide and Prestrud, 2005). The last day of the leaf senescence period marks the beginning of dormancy. The new phenology subroutine of CLM5-FruitTree also controls C reserve dynamics, stem and root turnover, and final rotation, which involves removing and replanting trees when the maximum orchard lifespan is reached.

### 2.2.2 Carbon and nitrogen allocation

CN allocation to the growth of new tissue (display pools) and to storage pools follows the phenological stages described in Sect. 2.2.1 (Fig. 2). A coupled CN allocation subroutine determines the fate of newly assimilated C from photosynthesis. A user-defined initial biomass can be assigned to leaf and fine root transfer pools via the *transplant* parameter (Table C1), while additionally 10 % of this biomass is assigned to the dead stem pool to define an initial stem area index >0. Each pool is also assigned the corresponding amount of N. Adjustments to this parameter have only little effect on the biomass growth and yield of the adult trees as the trees reach their maximum canopy height and develop their full leaf area index (LAI) within the first couple of years after transplanting. Thereafter, the potential allocation to the different plant components is based on allocation coefficients and allometric relationships between dead and live parts of stem and coarse

root. Throughout the growing period until harvest, 5 % of the newly assimilated C is allocated to the storage pools, as defined by the *fcur* parameter, except for fruits, where all allocated C is assigned to the displayed pool. For all other organs, the remaining C is also allocated to the displayed C pools. At bud break, a fraction of the C in the storage pool of all plant components, except fruits, is transferred to the actively growing C pools over a period that can be specified by the newly added parameter *ndays_stor*. This is based on the assumption that resources are partially mobilized to support growth of new tissue (Oliveira and Priestley, 1988; Loescher et al., 1990). Lacking more specific knowledge of the exact fraction, the default of 0.5 used by the seasonal deciduous phenology in CLM5 is adopted for fruit trees.

Before the start of fruit growth, phase (1), newly assimilated C and corresponding N are partitioned between leaf, stem, and root pools. The allocation coefficients are calculated according to a set of equations that were adapted from the AgroIBIS crop phenology algorithm used in CLM5–BGC-crop (Lawrence et al., 2018):

$$GDD_{T_{2\,m}} = GDD_{T_{2\,m}} + T_{2\,m} - T_f - 4,$$
$$\text{where } 0 \leq T_{2\,m} - T_f - 4 \leq 26 \text{ degree days}, \quad (1)$$

$$a_{repr} = 0, \quad (2)$$

$$a_{froot} = a_{froot}^i - \left(a_{froot}^i - a_{froot}^f\right) \times \frac{GDD_{T_{2\,m}} - GDD_{leaf}}{GDD_{fruit} - GDD_{leaf}}, \quad (3)$$

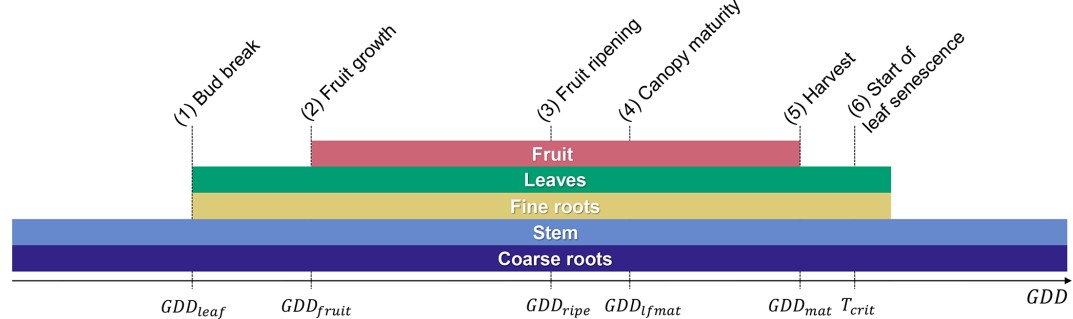

**Figure 2.** Fruit tree phenological stages of (1) bud break at the end of dormancy, (2) the start of fruit growth, (3) fruit ripening, (4) canopy maturity, (5) harvest, and (6) the start of leaf senescence. The lengths of phenological stages (2)–(5) are determined by their respective growing degree-day (GDD) thresholds starting from bud break (GDD$_{leaf}$ = 0), while stage (6) is determined by a critical temperature threshold ($T_{crit}$). Coloured bars correspond to the time any plant organ is present in the field throughout a year.

$$a_{leaf} = (1 - a_{froot}) \times \frac{a_{leaf}^{i} \times \left(e^{-b} - e^{-b \times \frac{GDD_{T_{2m}} - GDD_{leaf}}{GDD_{lfmat} - GDD_{leaf}}}\right)}{e^{-b} - 1}, \quad (4)$$

$$a_{livestem} = 1 - a_{repr} - a_{froot} - a_{leaf}, \quad (5)$$

where GDD$_{T_{2m}}$ are the accumulated growing degree days for the 2 m air temperature with maximum increments of 26 degree days, $T_{2m}$ is the simulated 2 m air temperature in Kelvin, $T_f$ is the freezing temperature of water and equals 273.15 K, GDD$_{leaf}$, GDD$_{fruit}$, and GDD$_{lfmat}$ are thermal thresholds for bud break, start of fruit growth, and canopy maturity, respectively, $b$ is an exponential factor, $a_{leaf}^{i}$, $a_{froot}^{i}$, and $a_{froot}^{f}$ are initial and final values for the allocation coefficients to leaf ($a_{leaf}$) and fine root ($a_{froot}$), respectively, and $a_{repr}$ and $a_{livestem}$ are the allocation coefficients to fruit and live stem, respectively.

Once fruit growth begins in phase (2), an increasing proportion of the assimilated C and corresponding N is allocated to this organ, causing leaf allocation to decline and fruit allocation to plateau at a high value once canopy maturity is reached. Allocation to fine roots and stem continues to decline and then settles at a constant value until harvest:

$$a_{livestem} = a_{livestem}$$

$$\times \left(1 - \frac{(GDD_{T_{2m}} - GDD_{leaf}) - (GDD_{fruit} - GDD_{leaf})}{(GDD_{mat} - GDD_{leaf})^{d_L - (GDD_{fruit} - GDD_{leaf})}}\right)^{d_{alloc}^{stem}}, \quad (6)$$

$$a_{repr} = 1 - a_{froot} - a_{livestem} - a_{leaf}, \quad (7)$$

where GDD$_{mat}$ is the thermal threshold for fruit maturity and harvest, while $d_L$ and $d_{alloc}^{stem}$ are stem allocation decline factors.

After harvest and until the start of dormancy, all of the newly assimilated C is sent to the storage pools following the notion that, late in the season, assimilates are used mostly to fill up reserves that can be mobilized to resume growth in the following spring (Le Roux et al., 2001). Fruit trees store C in the perennial woody parts of the tree, from where it is re-mobilized to support the growth of new shoots, leaves, and fine roots (Oliveira and Priestley, 1988; Millard, 1996;

Le Roux et al., 2001). Since in CLM5 separate storage pools are assigned to each plant organ, the newly added *aleafstor* parameter (Table C1) defines the fraction of allocatable C going to the leaf storage pool, while the remainder is split equally between roots and stem.

Fruit trees, similar to other deciduous species, have been observed to translocate N out of senescent leaves to be reused by other tree organs (Millard, 1996; Malaguti et al., 2001; Millard et al., 2006). Therefore, CLM5-FruitTree adopts the same N re-translocation strategy as used in the BDT phenology, during which N is removed from falling litter based on leaf and litter C : N ratios and the available C to pay for the extraction of N from increasingly recalcitrant litter pools. Subsequently, it is transferred to the plant N pool, from where it can be used for the growth of new plant tissue (Lawrence et al., 2018).

### 2.2.3 Representation of management practices

Furthermore, management practices such as fertilization and stem pruning are represented in the new sub-model. Fertilization is performed on a yearly basis after the occurrence of bud break, as N fertilization in early spring is still the most common practice in fruit orchards even though autumn fertilization or multiple applications via fertigation are also in use to increase fertilizer N use efficiency and reduce N losses (Sanchez et al., 1995; Carranca et al., 2018). We use the existing fertilization scheme of the crop phenology that adds fertilizer directly to the soil mineral N pool. A user-defined fertilization rate or amount can be applied as synthetic fertilizer or manure, respectively, although there currently is no difference in model behaviour for these two fertilizer types (Lawrence et al., 2018).

Winter pruning is a common practice in fruit orchards and may be performed throughout the winter to control the shape and size of fruit trees and partially to manage crop load (Grechi et al., 2008). In many intensive orchard production systems, pruning residues are mulched into the soil, possibly

increasing C sequestration (Montanaro et al., 2010; Aguilera et al., 2015). Alternatively, residues may also be exported and treated as waste (Benyei et al., 2018) or utilized for energy production (Kazimierski et al., 2021). In CLM5-FruitTree, pruning is performed as the tree enters dormancy by removing a user-defined fraction, *prune_fr* (Table C1), of the dead stem from both storage and displayed C pools. We remove C from the dead stem pool instead of the live stem pool since the former is the main wood pool in CLM5 that receives 85 % of the C allocated to total new wood. Furthermore, the implemented live wood turnover in CLM5 converts live stem to dead stem at the end of the growing season to account for differences in maintenance respiration and C : N ratios between these tissue types (Lawrence et al., 2018). Hence the live stem C pool remains rather small and stable over the years, so that applying pruning to this pool would have little effect on total tree biomass. The pruning implemented in CLM5-FruitTree affects only the tree biomass and height that are calculated based on this biomass pool, which in turn affects the calculation of turbulent fluxes of sensible and latent heat. However, this effect is small, and since turbulent fluxes are generally low in winter, the exact timing of pruning does not play a significant role in the magnitudes of these fluxes. During the first 3 years after planting, trees are not pruned to allow some initial stem biomass to grow. The sub-model treats pruning residues in one of two ways to account for their possible difference in fate: (1) residues are added to the wood harvest pool and exported from the field or (2) residues are added to the woody debris pool, thus feeding the litter cycle.

When the orchard reaches the end of its lifespan, C of all biomass pools (storage, transfer, and display) is sent to either the litter pool for leaves and fine roots or the wood harvest pool for live and dead stem and coarse roots, while any remaining C in the fruit pool is harvested. The orchard can then be replanted in the following year. Lastly, the standard irrigation routine implemented in CLM5 can be used for irrigated orchards by selecting the irrigated crop PFT.

## 2.3 Model implementation and testing

### 2.3.1 Site data

Extensive field measurements from an apple-growing region in the Adige River valley, South Tyrol, Italy (46°21′ N, 11°16′ E; 240 m a.s.l.) were used to parameterize and test the new CLM5-FruitTree sub-model along with the new apple PFT (Zanotelli et al., 2013, 2015, 2019). Measurements were obtained from an approximately 0.5 ha irrigated apple orchard planted in 2000 with the Fuji apple cultivar grafted on M9 dwarfing rootstock. The apple trees were planted at a row and tree spacing of $3 \times 1$ m (3333 trees per hectare). A 1.8 m-wide grass strip was grown between the tree rows, which was mowed three times a year. Other management practices included regular pruning, spring fertilization of 7.5 g N m$^{-2}$ yr$^{-1}$, and tillage of the soil directly un-

derneath the trees (Zanotelli et al., 2013). Stand-related data included general stand characteristics and phenology observations, LAI, C : N ratios, rooting distribution at three depth ranges (0–20, 20–40, and 40–60 cm), measurements of the biomass growth of different tree organs at a monthly or seasonal interval, and fruit harvest information (Table 1). Furthermore, daily soil respiration measurements from a control and a trenching plot (with ($R_s$) and without ($R_h$) root respiration, respectively) were performed in 2010. Additionally, an eddy covariance (EC) station provided measurements of the turbulent exchange of trace gases and energy at the studied apple orchard between 2013 and 2015. The quality check, gap filling, and flux partitioning of collected data followed the procedure outlined in Reichstein et al. (2005). The average closure of the energy balance was 60 %. To correct for the closure failure, the missing energy was assigned to the latent (LE) and sensible ($H$) heat fluxes based on the daily Bowen ratio (Zanotelli et al., 2019). Measured or derived fluxes included net ecosystem $CO_2$ exchange (NEE), ecosystem respiration ($R_{eco}$), gross primary production (GPP), LE, $H$, and evapotranspiration (ET) at half-hourly intervals. Furthermore, soil heat flux ($G$) measured at 5 cm depth as well as soil moisture measurements up to a depth of 60 cm of soil are available. Table 1 gives a summary of the available data and measurement periods. A complete description of the measurement procedures and instruments can be found in Zanotelli et al. (2013, 2015, 2019).

Meteorological data, recorded partly at the EC tower and at the Laimburg meteorological station located 4 km from the site (46°23′ N, 11°17′ E; 224 m a.s.l.), were used at an hourly time step to force the model. Measured data included precipitation, solar radiation, net radiation ($R_n$, only at the EC tower), air temperature, air pressure (only at Laimburg), relative humidity, and wind speed. Measurements of incoming longwave radiation (LW$_{in}$) were available for 2010 only, but additional calculations following Konzelmann et al. (1994) and Sedlar and Hock (2009) were produced and used as forcing for the remaining years 2011–2019 (Appendix B). This was necessary since the use of the internally calculated LW$_{in}$ in CLM5 resulted in unrealistic underestimations compared to the available measurements of LW$_{in}$, leading to a significant bias in $R_n$.

### 2.3.2 Model set-up

The model was set up in point mode to simulate the apple orchard in the Adige valley using available sand, clay, and organic matter fractions. The model was spun up for 200 years, first in accelerated decomposition and then in normal decomposition mode, until all state variables, such as total ecosystem soil C and soil water, reached equilibrium (Lawrence et al., 2018). For the model spin-up, the CRUNCEPv7 atmospheric forcing data set from 1986 to 2016 was used (Viovy, 2018). The apple orchard was then initiated using the newly developed sub-model and the apple PFT by selecting the site-

**Table 1.** Summary of available data from an apple orchard in the Adige River valley, South Tyrol, Italy, between 2010 and 2019. Solid lines represent continuous and dotted lines monthly measurements, while diamonds represent single measurements.

| Data | 2010 | 2011 | 2012 | 2013 | 2014 | 2015 | 2016 | 2017 | 2018 | 2019 |
|---|---|---|---|---|---|---|---|---|---|---|
| Weather data EC tower | ▬▬ | | | ▬▬▬▬ | | | | | | |
| Weather data Laimburg | ▬▬▬▬▬▬▬▬▬▬▬▬▬▬▬▬▬▬ | | | | | | | | | |
| Biomass (NPP) components | ••••• | ◆ | ◆ | | | | | | | |
| C:N ratios of biomass components | ••••• | | | | | | | | | |
| Leaf area index (LAI) | •••••••••••••••• | | | | | | | | | |
| Fruit production (yield) | | ◆ | ◆ | ◆ | ◆ | ◆ | ◆ | | | |
| Root distribution (0–60 cm) | ◆ | | | | | | | | | |
| Soil respiration (R$_s$, R$_h$) | ▬▬ | | | | | | | | | |
| Soil heat flux (G) | | | | ▬▬▬▬ | | | | | | |
| Soil water content (SWC, 0-60 cm) | | | | ▬▬▬▬ | | | | | | |
| EC data: carbon (GPP, R$_{eco}$, NEE), energy (R$_n$, LE, H), water (ET) | | | | ▬▬▬▬ | | | | | | |

specific management (i.e. fertilization with $7.5 \, \mathrm{gN \, m^{-2} \, yr^{-1}}$, irrigation, mulching of pruning material). Simulations were performed for a period of 10 years to mirror the time from orchard establishment in 2000 up to the start of the measure-
5 ments in 2010 using 10 years (2010–2019) of the available meteorological data from Laimburg meteorological station. Simulations were then extended for another 6 years from 2010 to 2015 for model parameterization and performance evaluation purposes.

### 2.3.3 Parameterization

Key parameters of the new sub-model as well as other PFT-specific parameters were parameterized using the first 3 years of simulations between 2010 and 2012. The lengths of phenological stages and associated parameters were determined
based on field observations of bud break, full bloom, and harvest as well as non-cultivar-specific apple phenology descriptions that were found in the literature (Appendix C). The length of the period where growth is supported out of reserves (*ndays_stor*) was calibrated based on the biomass
measurements and the estimate by Zanotelli et al. (2013) that apple trees use stored carbohydrates in the first 2 months after bud break. C allocation coefficients were calculated based on the monthly measurements in 2010 by dividing the biomass growth of the individual plant organs by the total biomass in-
crement. Subsequently, model parameters associated with the CN allocation subroutine (Eqs. 2–7) were calibrated manually to match the coefficients obtained from the observations and the overall biomass partitioning on a yearly basis. Parameter values for C : N ratios of all plant organs and max-
imum LAI were based on field observations in 2010 and 2010–2012, respectively. The specific LAI was calculated by dividing monthly measurements of LAI by leaf biomass and taking the average of the obtained values. Structural and morphological parameters such as maximum tree height

(*ztopmx*), planting density (*nstem*), the ratio of stem height to
35 radius at breast height (*taper*), or rooting depth (*root_dmx*) were adjusted based on site-specific information (Zanotelli et al., 2013). Initial biomass at transplanting was assumed to be $5 \, \mathrm{gC \, m^{-2}}$, resulting in an initial tree height of around 100 cm and a stem diameter of 16 mm. As seedlings are dor-
40 mant at the time of transplanting, their LAI is 0. The CLM5 root distribution parameter (*rootprof_beta*), which sets the root ratios at different depths, was calibrated by least squares regression of the measured root ratios at 0–20, 20–40, and 40–60 cm depths and the calculated ratios. Optical parame-
45 ters for leaf transmittance and reflectance in the visible and near infrared (IR) were set to average values reported for apple by Bastías and Corelli-Grappadelli (2012). Stem reflectance and transmittance were assumed to be similar to other woody species and therefore set to the values used for
BDT in CLM5, similar to the assumptions made by Fan et al. (2015) for the palm tree development in CLM4.5. The ratio of momentum roughness length to canopy top height (*z0mr*) was set to the average value of the ranges reported for apple and citrus orchards to account for the differences in
canopy structure compared to annual crops and forest (Tanny and Cohen, 2003; de la Fuente-Sáiz et al., 2017). No specific values could be found for the ratio of displacement to top of canopy height (*displar*), the leaf orientation index (*xl*), or the intercept to calculate the top of canopy maintenance res-
piration base rate (*lmr_intercept_atkin*). These values were assumed to be comparable to other deciduous trees and thus set to the values used for BDT in CLM5. Parameters related to C reserve dynamics (e.g. *fcur*) and photosynthesis (e.g. the slope of the relationship between leaf N per unit area and
the maximum rate of carboxylation at $25\,°C$, *s_vcad*) were adjusted to match observed LAI and productivity data. All parameters with their values and references to the literature are summarized in Table C1 of the Appendix.

### 2.3.4 Sensitivity analysis

A simple one-by-one sensitivity analysis was performed to further tune model parameters and assess the influence of newly added parameters on the simulation results. As a complete sensitivity analysis of all PFT-related parameters would have exceeded the scope of this study, the analysis focused on key parameters of the new phenology and CN allocation subroutines. Other potentially influential parameters were selected based on previously performed sensitivity analyses by Göhler et al. (2013) for CLM3.5 and by Cheng et al. (2020) and Dagon et al. (2020) for CLM5, taking into account differences between previous and current model versions. Parameters selected for the analysis were perturbed by varying a parameter by $\pm 30\%$, $\pm 20\%$, and $\pm 10\%$ while keeping the others fixed to the value of the control simulation (after initial parameterization). The goal here was not to perform an in-depth analysis covering the full range of possible parameter values but rather to provide a first indication of influential parameters in the new sub-model similar to the approach of Fan et al. (2015). As a measure of sensitivity, the parameter effect (PE) was calculated using the average of three years of simulations between 2013 and 2015 of the control and the perturbed simulations for selected output variables and the following formula adjusted from Luo et al. (2020):

$$\Delta X_{i,j} = \sum_{k=1}^{n} \frac{\left| \overline{X_{i,j,k}} - \overline{X_{i,\text{control}}} \right|}{\left| \overline{X_{i,\text{control}}} \right|}, \tag{8}$$

$$\text{PE}_{i,j} = \frac{\Delta X_{i,j}}{max\left[ \left( \Delta X_{i,j} \right)_{1 \le i \le n; 1 \le j \le m} \right]}, \tag{9}$$

where $X$ is a simulated value of the control or a perturbation run, $\Delta X$ is the summed absolute difference between the control and the perturbation run across all perturbations, $k$ is the parameter perturbation factor, $i$ is the $i$th variable across $n = 6$ selected output variables including GPP, NEE, $R_a$, LE, maximum LAI, and yield, and $j$ is the $j$th parameter across $m$ selected parameters. $\text{PE}_{i,j}$ is a number between 0 and 1 that represents the sensitivity of an output variable $i$ to the parameter $j$, with 1 meaning high and 0 meaning low sensitivity. The parameters selected for sensitivity analysis are indicated in Table C1 of the Appendix.

### 2.3.5 Model performance evaluation

Modelling results are compared to observed biomass, yield, and LAI data as well as ecosystem fluxes retrieved from the EC measurements. Statistical indices for model performance evaluation include the Pearson coefficient of correlation ($r$), the root mean square error (RMSE), and the percent bias error (%bias):

$$r = \frac{\left( \frac{1}{n} \sum_{i=1}^{n} \left( X_i^o - \mu^o \right) \times (X_i - \mu) \right)}{\sigma \times \sigma^o}, \tag{10}$$

$$\text{RMSE} = \sqrt{\frac{1}{n} \sum_{i=1}^{n} (X_i - X_i^o)^2}, \tag{11}$$

$$\%\text{bias} = \frac{\sum_{i=1}^{n} \left( X_i - X_i^o \right)}{\sum_{i=1}^{n} \left( X_i^o \right)}, \tag{12}$$

where $i$ is the time step, $n$ is the total number of time steps, $X_i$ and $X_i^o$ are simulated and observed values at each time step, respectively, $\mu$ and $\mu^o$ are simulated and observed mean values, respectively, and $\sigma$ and $\sigma^o$ are simulated and observed standard deviations.

## 3 Results and discussion

### 3.1 Sensitivity analysis

A total of 34 parameters were initially considered for the sensitivity analysis, of which the 13 most influential parameters (PE $>0.1$ for at least one of the selected output variables) are shown in Fig. 3. GPP, NEE, $R_a$, and yield have similar sensitivity patterns and are most sensitive to the leaf C : N ratio (*leafcn*) and the relationship between leaf N and the maximum rate of carboxylation at 25 °C (*s_vcad*). Together with the specific leaf area (*slatop*) and other constants, they control the maximum photosynthetic capacity in the photosynthesis calculation and thus largely influence total C assimilation. As expected, LAI is most influenced by parameters that control the CN allocation to leaves such as the initial leaf allocation coefficient (*fleafi*), the GDDs needed to reach canopy maturity (*lfmat*), the maximum LAI (*laimx*), photosynthetic parameters, and, to a smaller extent, the fraction of C allocated to the leaf storage pool to refill C reserves (*aleafstor*). The first three parameters influence leaf biomass and thus show a considerable effect on GPP, NEE, $R_a$, and yield. The same output variables are affected in a similar fashion by the GDDs needed until fruit harvest (*hybgdd*) that control the amount of C allocated to fruits. LE is influenced largely by the parameter controlling stomatal conductance (*medlynslope*) and the photosynthetic parameters (*leafcn*, *s_vcad*).

Overall, photosynthetic parameters play a key role in determining the magnitude of the studied output variables, with an average PE value close to 0.7 across all six variables. Phenological parameters (top seven parameters in Fig. 3) are generally less influential for the same output variables, with average PE values up to 0.43. These findings are largely consistent with earlier studies of parameter sensitivity (Göhler et al., 2013; Cheng et al., 2020; Dagon et al., 2020; Luo et al., 2020). In contrast to Luo et al. (2020), we did not find a strong effect of the root distribution parameter (*rootprof_beta*) on LE, which can be attributed to the low water stress due to the irrigation management of the studied orchard.

While the one-at-a-time sensitivity analysis provides some insight into model sensitivity, the ranking of influential parameters is strongly influenced by the choice of parameters

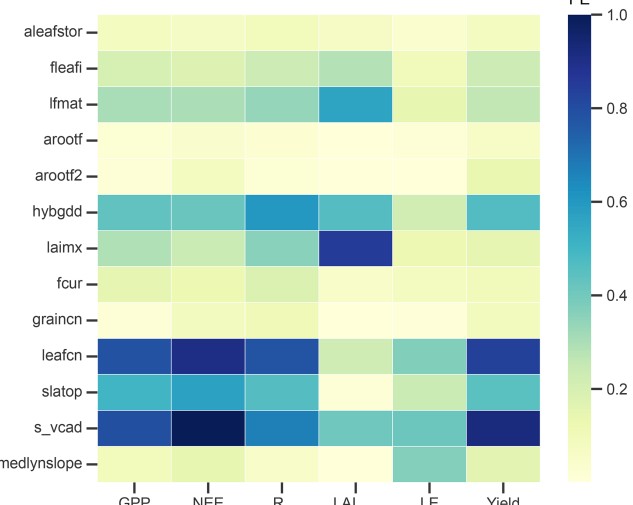

**Figure 3.** Parameter effect (PE) as a measure of sensitivity of selected output variables to the most influential model parameters. Output variables include gross primary production (GPP), net ecosystem exchange (NEE), autotrophic respiration ($R_a$), maximum leaf area index ($LAI_{max}$), latent heat flux (LE), and yield. Parameters are post-harvest leaf allocation coefficient to storage (*aleafstor*), initial leaf allocation coefficient (*fleafi*), GDD to canopy maturity (*lfmat*), root allocation coefficients at the start of fruit development (*arootf*) and until harvest (*arootf2*), GDD needed until harvest (*hybgdd*), maximum LAI (*laimx*), fraction of allocation that goes to currently displayed growth (*fcur*), C : N ratios of fruits (*graincn*) and leaves (*leafcn*), specific leaf area at top of canopy (*slatop*), slope of the relationship between leaf N per unit area and the maximum rate of carboxylation at 25 °C (*s_vcad*), and the medlyn slope of the conductance–photosynthesis relationship (*medlynslope*). For more details on the parameters, see Appendix C.

and output variables, the parameter perturbation strategy (i.e. percent change, linear sampling), and the index chosen as the sensitivity measure. Parameter tuning based on this analysis is further complicated since this approach does not consider parameter covariation that is particularly strong for plant parameters that influence photosynthesis (Göhler et al., 2013). Selecting parameter values based on the individual best simulation hence does not necessarily yield the best overall result (Luo et al., 2020). We therefore decided to first adjust *s_vcad* to best match the observed average GPP. In the following, we further adjusted *fleafi*, *hybgdd*, and *medlynslope* to improve the simulated biomass components as well as the LE flux, respectively.

## 3.2 Modelling results

In the following, we present the modelling results according to the initial parameterization and the updated parameter values from the sensitivity analysis. Daily simulations or yearly sums are compared to observed biomass, yield, and LAI data as well as ecosystem fluxes retrieved from the EC measurements and soil moisture measurements aggregated to daily mean values.

### 3.2.1 Biomass growth and yield

The patterns in seasonal biomass allocation simulated by CLM5-FruitTree show good agreement with the monthly observations from 2010 (Fig. 4a). The beginning and end of the growing season are well captured. After bud break at the beginning of March, biomass is allocated to the vegetative organs of leaves, fine roots, and woody organs, and growth is supported by C and N reserves until the start of fruit growth in early May (50 d according to the *ndays_stor* parameter). In the following months, fruit biomass grows rapidly until harvest takes place in mid-October, following the typical sigmoidal growth curve that is well captured by the new phenology and CN allocation. Simulated leaf biomass peaks in mid-June and remains constant thereafter, with leaf senescence starting later in October when temperatures drop below 4 °C. Pruning is performed when the tree enters dormancy by removing 85 % of the stem biomass assimilated over the season according to the observed pruning amounts in the studied apple orchard (Zanotelli et al., 2013, 2015). From 2010 to 2012, the modelled percentage of biomass allocation to plant organs was generally in agreement with the observations (Zanotelli et al., 2015), with differences ranging between 1 % and 5 % for fruits, leaves, aboveground wood, and roots (Fig. 4b). Penzel et al. (2020) stated that different studies reported biomass allocation to fruits ranging from 50 % to 85 % depending on apple cultivar, suggesting considerable variability in allocation coefficients. This emphasizes the benefit of a cultivar-specific calibration in order to obtain realistic modelling results. On the other hand, it suggests that a more general parameterization that reflects an average apple tree may be necessary to apply CLM5-FruitTree at larger scales and across multiple cultivars.

The timing for initial leaf development in spring and leaf senescence in late autumn are sufficiently well captured by the implemented bud break prediction algorithm and the simple temperature threshold for leaf abscission, respectively (Fig. 5). Observed maximum LAI varied between 2.8 and 3.3 m$^2$ m$^{-2}$ and occurred during the first half of July. The simulations reached similar values in 2010 and 2012, matching the observations, while the simulated LAI in 2011 underestimated the measurements due to a smaller C transfer from storage and lower solar radiation early in the growing season. The discrepancy between the low simulated LAI and the high observed LAI in 2011 could have been further exacerbated by a lighter pruning performed in the previous winter compared to other years (Zanotelli et al., 2013). Such practice is sometimes performed in an attempt to counteract the strong alternate bearing behaviour of the Fuji variety, which causes a substantial drop in yield following a high yielding year (Belleggia et al., 2009; Atay et al., 2013; Pasa et al., 2021). As a consequence of the light pruning, a larger num-

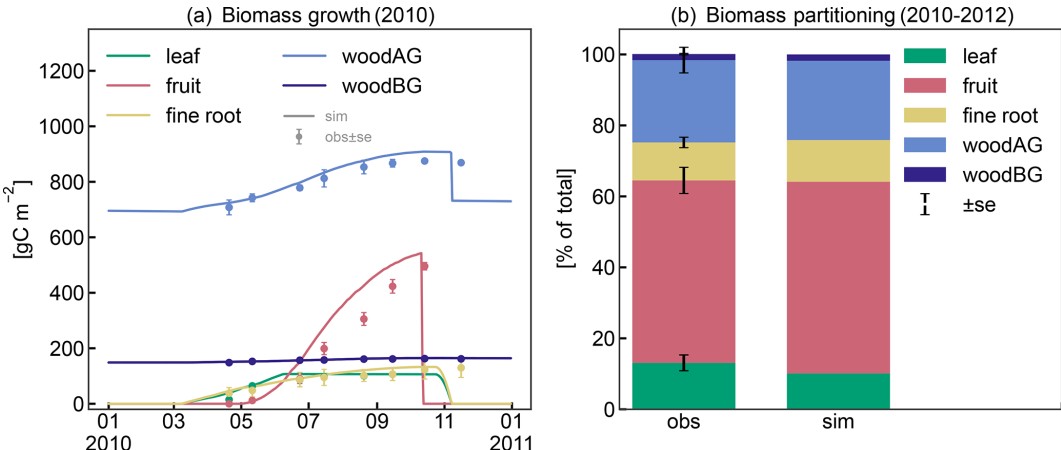

**Figure 4. (a)** Observed and simulated growth of leaves, fruits, fine roots, and aboveground (live and dead stem) and belowground (live and dead coarse roots) biomass during 2010. **(b)** Observed and simulated biomass components between 2010 and 2012 as percentage of total biomass.

ber of vegetative and flower buds remained on the tree, leading to more growth and possibly contributing to the larger discrepancy between relatively high observed LAI and relatively low simulated LAI. The adjusted pruning is however
based on a somewhat subjective assessment of the farmer, and information about the exact amount is hardly available. Thus, CLM5-FruitTree currently adopts a simplified pruning practice based on the removal of a fixed portion of the seasonal stem growth which manages tree size and total woody
biomass without affecting LAI.

Measured LAI showed a slow decline soon after maximum LAI was reached, while simulated values in contrast are assumed to remain constant until leaf senescence is initiated. The observed early decline may be an artefact of the sam-
15 pling strategy used to determine LAI that extrapolated individual leaf area measurements to the whole tree, assuming a constant leaf distribution within the tree (Zanotelli et al., 2013). Another reason could be some premature leaf fall in the summer at the expense of the inner shadowed leaves, as
observed during field sampling. Other studies suggest that the LAI of fruit trees generally stays constant until a rapid decline with the start of senescence (Lakso et al., 1999; Pallas et al., 2016), supporting the simulated LAI dynamic.

Simulated yield averaged $70\,\mathrm{t\,ha^{-1}}$ between 2010 and
25 2015 and was within 2.3 % of the observed average yield. While simulated yield varied between 61 and $76\,\mathrm{t\,ha^{-1}}$, the observations showed a greater inter-annual variability (IAV), as exemplified in the case of the years 2012 (low yield of $51\,\mathrm{t\,ha^{-1}}$) and 2015 (high yield of $101\,\mathrm{t\,ha^{-1}}$) (Fig. 6). Low
IAV of yield has also been observed in previous crop simulations with CLM5 for winter wheat (Boas et al., 2021), suggesting that certain drivers of IAV such as extreme environmental conditions (e.g. frost, heat, and hail) or plant pests and diseases and the resulting plant physiological responses
(e.g. stress-induced leaf shedding or failure to flower) (Char-

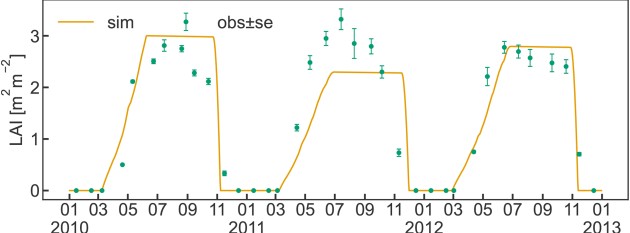

**Figure 5.** Simulated daily leaf area index (LAI) between 2010 and 2012 together with observations (± standard error) of LAI that were made once a month for the same period. Ticks on the *x* axis refer to the beginning of the month.

rier et al., 2021) are missing or not represented with sufficient detail in CLM5. In the case of apple trees, yield is also tightly linked to the number of flowers and early fruit growth, which in turn depends on a complex interaction of the environmental conditions during winter dormancy and the start of
40 the new growing season (Chmielewski et al., 2012; Corelli-Grappadelli and Lakso, 2002). Additionally, C reserves accumulated in the previous year (Greer et al., 2002), and crop load management played an important role in determining the final harvest (Penzel et al., 2020). The latter includes
pruning or fruit thinning to ensure optimal fruit growth and to reduce the effect of alternate bearing. The low observed yield in 2012 may be a result of such behaviour. This phenomenon and the processes involved are not universal, so that different fruit trees may be bearing regularly, irregularly, or biannu-
ally (Hoblyn et al., 1937; Monselise and Goldschmidt, 1982). As such, alternate bearing and its treatment through pruning or fruit thinning cannot easily be generalized and are thus not currently implemented in CLM5-FruitTree, which could have further reduced simulated IAV. Storage growth is con-
sidered in CLM5-FruitTree and exhibited an impact on the

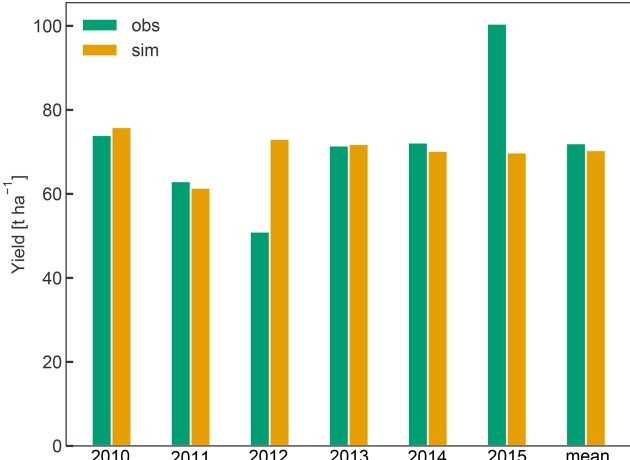

**Figure 6.** Annual yields from 2010 to 2015 and their mean in tons of fresh weight per hectare. For the conversion of simulated fruit biomass in gram carbon per square metre to tons per hectare, fruit C content was assumed to be 42 % of total dry weight, harvest efficiency 95 %, and fruit water content 83 % according to Zanotelli et al. (2013).

final yield of the following season, as shown by the sensitivity analysis of the *aleafstor* and *fcur* parameters (Fig. 3). However, its effect on fruit growth in CLM5-FruitTree is indirect since it supports leaf development in the early growth stage but does not directly contribute to fruit growth. Identifying the driving forces of reserve deposition and mobilization and their quantification remains an unsolved issue, and there is still no consistent formulation of this process in tree modelling (Le Roux et al., 2001; Allen et al., 2005). Predicting final yield in fruit orchards is further complicated by the fact that harvest is usually based on certain fruit quality traits such as firmness or soluble solids and can occur successively as fruits may not mature at the same time (Corelli-Grappadelli and Lakso, 2002; Musacchi and Serra, 2018). Within this context, the proposed simplifications of the C reserve dynamics and fruit harvest are likely contributing to the difference in observed and simulated yields. Considering the many specific challenges in modelling this apple cultivar, we believe that the yield predictions are satisfactory enough in the context of the sub-model development.

### 3.2.2    Ecosystem fluxes and soil moisture variation

#### Carbon fluxes

As shown in Fig. 7, CLM5-FruitTree was able to capture the overall patterns of GPP, NEE, and $R_{eco}$, particularly during the transition between dormancy periods and growing seasons (April to November). Simulated C fluxes are highly correlated with observations ($r \geq 0.84$), while the RMSE ranges between 1.12 and 1.53 gC m$^{-2}$ d$^{-1}$. Observed and simulated peak C fixation occurred in mid-June (Fig. 7a), correspond-

ing to the maximum (negative) NEE (Fig. 7c) and maximum LAI (Fig. 5). Simulated NEE becomes negative (net carbon sink) around April and returns to positive (net carbon source) around November, in agreement with the observed dynamic (Fig. 7c). Observed yearly sums of GPP (NEE) were 1.60 ($-0.49$), 1.43 ($-0.48$), and 1.65 ($-0.76$) kgC m$^{-2}$ yr$^{-1}$ for 2013, 2014, and 2015, respectively. Simulated yearly sums of GPP (NEE) were 1.58 ($-0.53$), 1.56 ($-0.51$), and 1.53 ($-0.57$) kgC m$^{-2}$ yr$^{-1}$ for the same years, showing a negligible positive bias of on average 0.17 % for GPP (Fig. 7b) and a small underestimation (less negative) of on average 3.8 % for NEE (Fig. 7d). Simulated and observed $R_{eco}$ (Fig. 7e) generally increased until July because of the increase in air temperature and respiratory costs of the developing canopy and declined thereafter as air temperature started to drop. Simulations of $R_{eco}$ tend to slightly underestimate observations between April and late August and to overestimate observations during winter, although discrepancies are relatively small. Observed yearly sums of $R_{eco}$ were 1.13 (2013), 0.98 (2014), and 0.94 (2015) kgC m$^{-2}$ yr$^{-1}$, while simulated values were 1.08, 1.08, and 0.99 kgC m$^{-2}$ yr$^{-1}$, respectively. CLM5-FruitTree overestimated yearly $R_{eco}$ by on average 3.3 %, explaining most of the difference in observed and simulated NEE in 2013, while differences in 2014 and 2015 are due to a combination of small biases in both GPP and $R_{eco}$. Measured $R_{eco}$ showed irregular fluctuations in the early part of the growing season 2013 and mid to late season 2014 and 2015 that are not reproduced well by the model. These fluctuations mostly correspond to the observed temperature dynamics (not shown) as a result of the applied gap filling that is based on an air (or soil) temperature–$R_{eco}$ relationship (Reichstein et al., 2005). Such discrepancies between observed and simulated dynamics could be further explained by the occurrence of field management practices such as mowing of the grassed alleys or soil tillage under the tree rows, which are currently not represented in CLM5-FruitTree. Such practices could have led to a temporary rise in soil respiration ($R_s$) due to increased heterotrophic respiration ($R_h$) as discussed in Zanotelli et al. (2013). Indeed, soil tillage experiments performed in an apple orchard located on the Loess Plateau in Shaanxi Province in China were found to increase $R_s$ by 14 %–57 % depending on the tillage method (Hou et al., 2021).

Zanotelli et al. (2013) measured a total $R_s$ of $801 \pm 95$ gC m$^{-2}$ in 2010, contributing around 90 % to $R_{eco}$, based on soil chamber measurements within the orchard (total soil respiration). The comparison to parallel measurements in a trenched plot produced a high ratio $R_h/R_s$ of 0.77 for the apple orchard. In contrast, simulated $R_s$ was 510 gC m$^{-2}$, contributing merely 45 % to $R_{eco}$ for the same year, with a ratio $R_h/R_s$ of 0.87. Simulated $R_{eco}$ was instead dominated by autotrophic respiration ($R_a$) due to high C costs for maintenance, mainly of leaf biomass (data not shown). Other studies found that $R_s$ contributed 56 %–67 % to $R_{eco}$ in irrigated citrus orchards of different

ages that share common management practices (i.e. use of heavy machinery, irrigation, fertilization, tree pruning, and mulching) as well as structural similarities (e.g. planting in tree rows) with the studied apple orchard. Both aspects have a strong influence on soil respiration components in orchards (Martin-Gorriz et al., 2020). In forest ecosystems, where the magnitude of ecosystem fluxes was found to be somewhat comparable to orchards, $R_s$ contributed $>60\%$ to $R_{eco}$ (Lasslop et al., 2012; Zanotelli et al., 2013).

In addition to the missing representation of certain management practices, CLM5-FruitTree currently does not account for an active ground cover in the orchard, which has been shown to enhance $R_s$ in an Italian olive orchard through increased fine root and microbial biomass (Turrini et al., 2017). Furthermore, the simplified representation of microbial activity in CLM5, through fixed respiration fractions for litter and soil organic matter pools, may limit the ability of CLM5-FruitTree to accurately represent soil respiration processes. Not accounting for mycorrhizal respiration may fail to adequately represent $R_{eco}$ of the orchard, as measurements suggested a substantial contribution of $11 \pm 6\%$ to total $R_s$ in an apple orchard (Tomè et al., 2016). Lastly, biases in simulated soil temperature, soil moisture content, and fine root density could further contribute to explaining the above-discussed differences, as these factors have a major effect on $R_s$ in apple orchards (Ceccon et al., 2011).

In contrast to the underestimation of $R_s$ in the model, the simulated $R_a$ of $693\,\mathrm{gC\,m^{-2}}$ was almost twice the measured value of $372 \pm 195\,\mathrm{gC\,m^{-2}}$. In our simulations, maintenance respiration comprised the main part of $R_a$, with on average 78 %. The calculation of maintenance respiration in CLM5 (see Sect. 2.1) does not account for a lower or varying maintenance cost observed in mature apple orchard canopies compared to annual crops (Bepete and Lakso, 1996; Lakso et al., 1999). It therefore seems likely that the tissue maintenance costs in the orchard are overestimated in CLM5-FruitTree, accounting for on average 45 % of $R_a$ (28 % of $R_{eco}$). This could also explain the lower simulated carbon use efficiency (NPP/GPP) of 0.59 compared to 0.71 found by Zanotelli et al. (2013). Further work and more experimental data are needed to better understand the differences in modelled and observed respiration partitioning and to improve the performance of CLM5-FruitTree to adequately simulate the respiration components in fruit orchards.

**Energy and water fluxes**

The simulated seasonal course of the energy balance components $R_n$, $G$, LE, and $H$ agrees well with observed dynamics in the orchard (Fig. 8). CLM5-FruitTree shows a high performance in reproducing $R_n$ and LE, with $r \geq 0.97$ and RMSE of 15.98 and $17.85\,\mathrm{W\,m^{-2}}$, respectively (Fig. 8a and c). Due to the lack of $LW_{in}$ measurements, the CLM5-internal $LW_{in}$ calculation based on a clear-sky parameterization after Idso (1981) was used initially. This resulted in a significant underestimation of 5 % (511 MJ) for $LW_{in}$ and 18 % (471 MJ) for $R_n$ compared to the observations in 2010. The $R_n$ bias could be reduced by 14 % for the observed time series when $LW_{in}$ was calculated by considering cloud cover as described in Appendix B. This stresses the necessity of accounting for cloud cover, ideally combined with locally calibrated parameters, for an accurate calculation of $LW_{in}$. The remaining small negative bias of 4.48 % in $R_n$ is due to negative simulated $R_n$ during the winter months (Fig. 8b), which may be a result of the higher reflectance of solar radiation from bare soil compared to a grass surface (Bryś et al., 2019). The model assumes a bare soil (except for stem area) during the dormancy period, as the grass-covered alleys in the orchard are not considered explicitly.

The simulated LE (Fig. 8c) shows similar dynamics and variability to the observations following the increase and decrease in GPP (Fig. 7a) and LAI (Fig. 5). Similarly to LE, modelled ET shows a high correlation coefficient of 0.97 and a small RMSE of $0.62\,\mathrm{mm\,d^{-1}}$ (Fig. 8i). Simulated ET exceeds observed ET by $1.1\,\mathrm{mm\,d^{-1}}$ on average during its peak in July, but the overall bias is almost negligible (Fig. 8j). Total observed ET is 901 (2013), 858 (2014), and 883 (2015) mm, while the corresponding simulated values are 916, 877, and 925 mm, respectively. When examining the order of magnitudes of the ET components, canopy transpiration takes up around 85 % of ET, followed by soil evaporation and canopy evaporation (data not shown). Typically, apple orchard ET represents a combined flux from the apple trees and the grassed alley system, which is not explicitly represented in CLM5-FruitTree since CLM5 currently does not consider inter-row grass coverage or intercropping. Ntshidi et al. (2021) found that the contribution of understory transpiration is high in young, non-bearing apple orchards but contributes less than 10 % to whole-orchard ET in mature orchards with high canopy cover, which may explain the good model performance despite not considering the grass cover.

Simulated $H$ and $G$ are less consistent with the observations, with $r$ values of 0.54 and 0.64, respectively, and large percent bias (Fig. 8e and g), which is partially due to the much smaller magnitudes of the two fluxes compared to $R_n$ and LE. A possible reason for the lower amplitude of observed $G$ (Fig. 8h) compared to simulated values may be the dampening effect of the grass cover providing additional shading during summer and insolation during winter (Bryś et al., 2019; Oorthuis et al., 2021). Observed $H$ was rather constant throughout the year, with slightly higher values at the start and end of the growing season when the canopy was not yet fully developed or leaves were shedding. CLM5-FruitTree simulated a clear rise of $H$ until April, closely following the observations, but $H$ thereafter declined steeply in May, with negative values in August 2013 and 2015. Negative $H$ during August corresponds to maximum LE and the main simulated irrigation season (June to September) that added 357 (2013), 281 (2014), and 517 mm (2015) of water to the orchard (Fig. 9a). In a study conducted with CLM4.5,

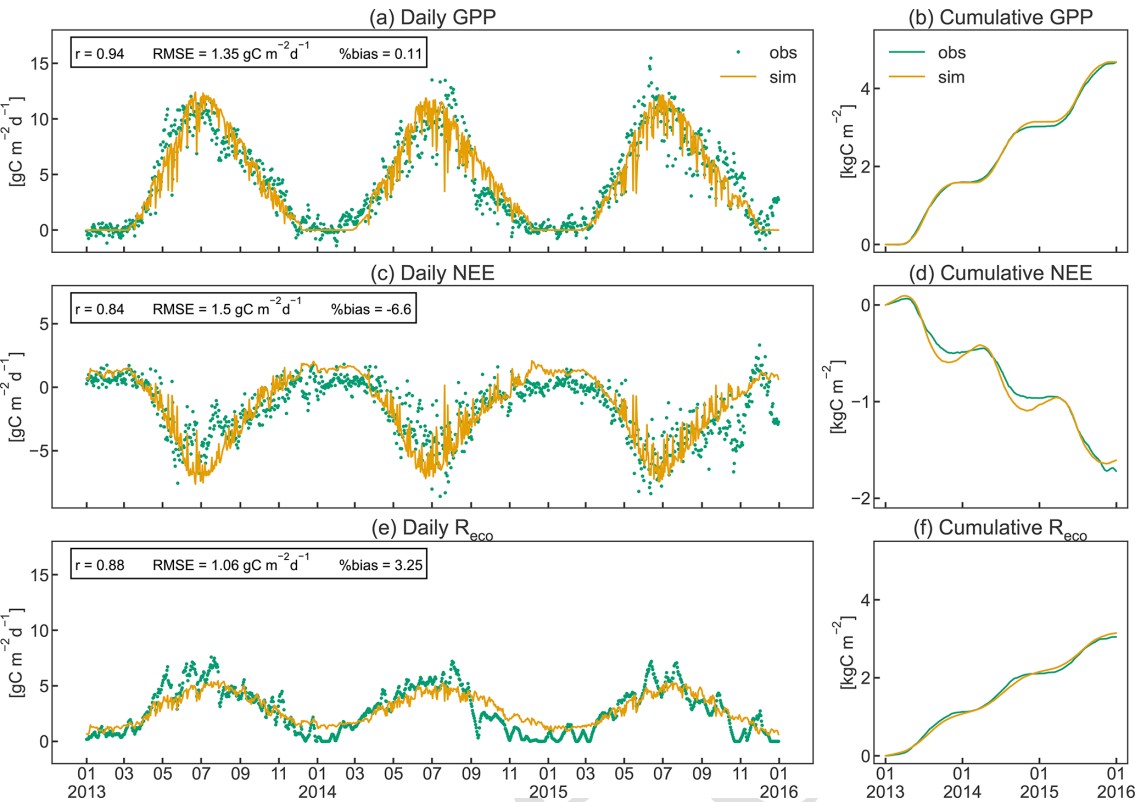

**Figure 7.** Daily instantaneous **(a, c, e)** and cumulative **(b, d, f)** observed and simulated fluxes of gross primary production (GPP), net ecosystem exchange (NEE), and ecosystem respiration ($R_{eco}$) for the studied apple orchard between 2013 and 2015. Pearson's coefficient of correlation ($r$), the root mean square error (RMSE), and the percent bias (%bias) are displayed as statistical indices.

intense irrigation was found to strongly influence the convective heat fluxes by increasing LE and decreasing $H$ (Zeng et al., 2017). Although precise measurements of the irrigation amount in the orchard are not available for the studied period, the average yearly irrigation was estimated around 200 mm, with no irrigation in 2014 due to sufficient rainfall (Montagnani et al., 2018). The difference in irrigation amounts may in part explain why the described phenomenon is not observed in the measurements. Indeed, negative simulated $H$ in the summer months occurred as a result of strong evaporative cooling of ground and vegetation temperature through energy absorption by LE following irrigation that caused simulated LE to exceed simulated $R_n$. This behaviour was not observed in the measurements where LE rarely exceeded $R_n$ and was mostly due to an overestimation of simulated LE compared to the measurements. Persisting model weaknesses in the partitioning of the energy balance were pointed out by a recent study examining land surface processes over a tropical rainforest using CLM4.5 and CLM5 and were linked to missing detail in the representation of the canopy and an oversensitivity of vegetation temperature to incoming solar radiation, among others (Song et al., 2020). As a result, the authors observed an overestimation of LE and unrealistically high day-to-night changes in $G$, which was also observed in this study

when examining the model output at an hourly time step (results not shown).

Energy partitioning in orchards is strongly influenced by the positioning and pruning of branches to optimize tree architecture for higher productivity, planting density, tree height, and LAI distribution (López-Olivari et al., 2016). Consequently, the contribution of $H$ and LE can significantly differ in the discontinuous orchard canopy (grass-covered alleys between tree rows) compared to the closed canopies of annual crops (de la Fuente-Sáiz et al., 2017). Currently CLM5 is still limited to the assumption of a closed canopy structure that is uniform in space, and hence biases are likely to arise from this model limitation. Future developments towards integrating multi-layer schemes for canopy processes and the explicit representation of the canopy to improve the related processes are desirable for a more realistic representation of the orchard canopy structure.

**Soil moisture variation**

Simulated mean soil moisture (SM) at 5 cm depth was within 1.6 vol % of the observed value during the three observed growing seasons, despite the higher simulated irrigation amount (Fig. 9b). Simulated daily values show a greater variability than the measured data in response to precipitation

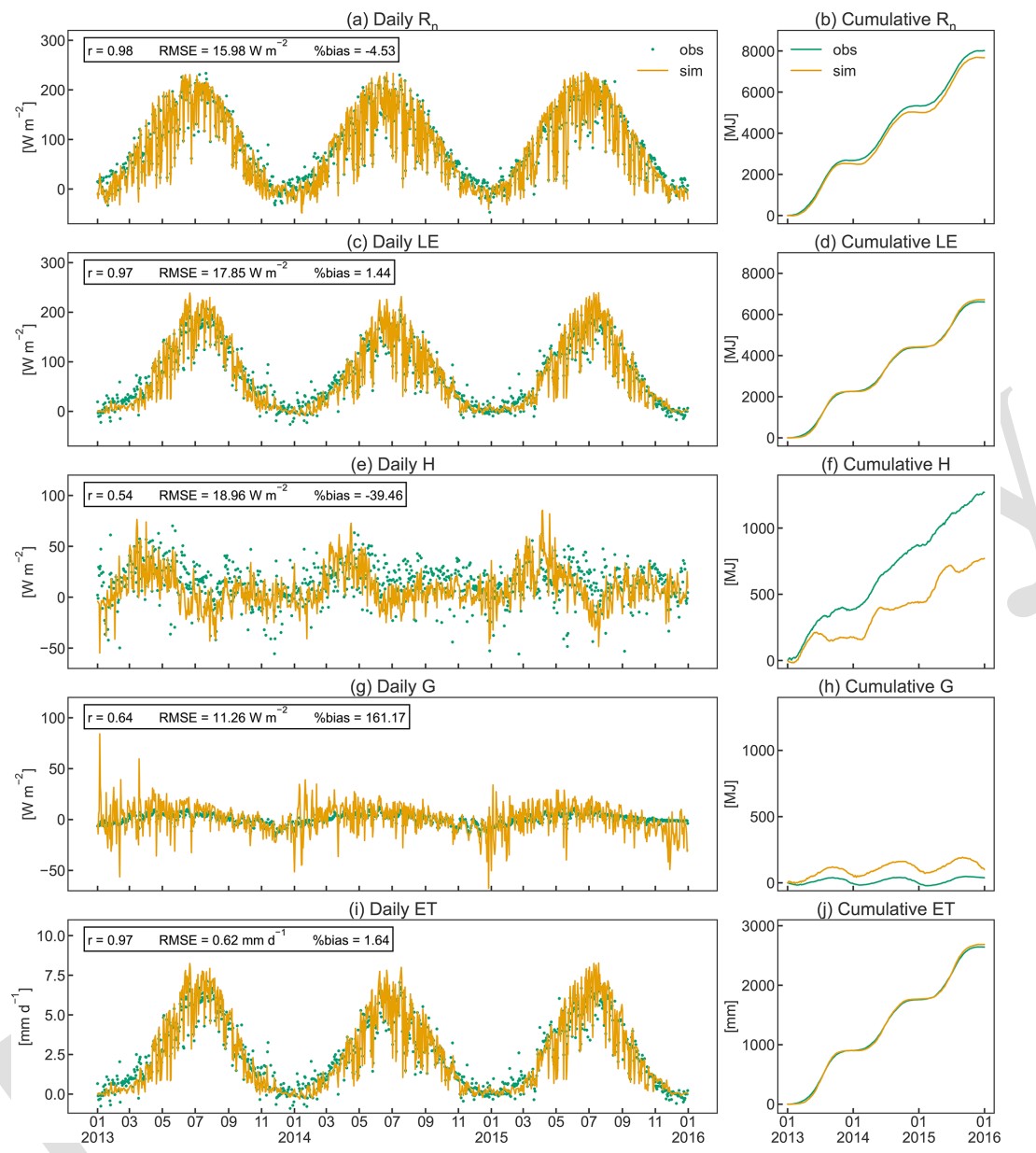

**Figure 8.** Daily **(a, c, e, g, i)** and cumulative **(b, d, f, h, j)** observed and simulated fluxes of net radiation ($R_n$), soil heat ($G$), latent heat ($H$), sensible heat (LE), and evapotranspiration (ET) for the studied apple orchard between 2013 and 2015. Pearson's coefficient of determination ($r$), the root mean square error (RMSE), and the percent bias (%bias) are displayed as statistical indices.

and to frequent irrigation (Fig. 9a–b). In contrast, observed SM in the deeper soils (30–60 cm) was 3 vol %–11 vol % higher during the growing season compared to simulated values (Fig. 9c–d). Considering the total investigated soil depth, simulations exhibit a larger variability in SM throughout the year, with a general overestimation in winter and underestimation during the growing season (especially in the deeper soils). However, the collected SM data were limited to a single soil profile that may not adequately reflect the average soil moisture of the apple orchard, which should be considered when comparing measurements and simulations. Even

though the measurements are incomplete, the constant high observed SM in the deeper soils suggests an ample supply of water due to capillary rise from the shallow groundwater table that typically ranges between 1.2 and 1.85 m in the area (Montagnani et al., 2018). This process replenishes the water removed by ET processes and may explain the reduced need for irrigation compared to the simulations. Despite the shallow simulated groundwater table (generally 1.2 m depth), groundwater could not be used for root water uptake in the simulation as the rooting depth of the orchard was restricted

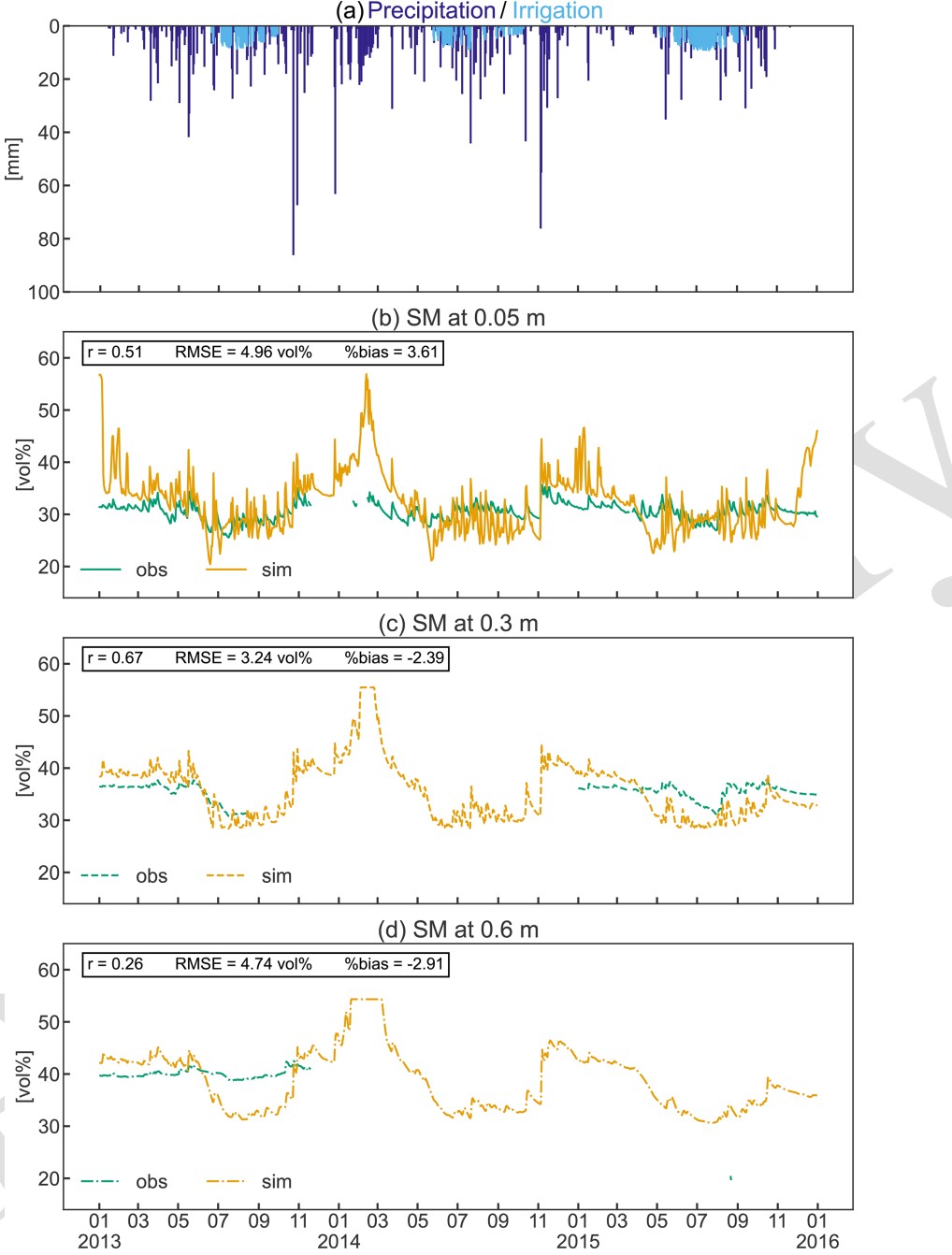

**Figure 9.** Precipitation and simulated irrigation **(a)** and observed and simulated soil moisture (SM) at 0.05 m **(b)**, 0.3 m **(c)**, and 0.6 m **(d)** depths from 2013 to 2015. Pearson's coefficient of correlation (*r*), the root mean square error (RMSE), and the percent bias (%bias) are displayed as statistical indices. TS2

to 0.8 m according to local measurements, and capillary rise is currently not implemented in CLM5.

## 4   Conclusions

The novel CLM5-FruitTree was developed to model perennial deciduous fruit orchards and thus extended the repre-

sentation of agricultural systems in CLM5. The development included a new phenology subroutine to account for the perennial nature, prolonged growing season, and distinct phenological development of fruit trees compared to annual crops. Furthermore, C reserve dynamics of perennial deciduous trees were considered by adapting the CN allocation, and typical management practices associated with fruit orchards

were represented, such as transplanting of seedlings and winter pruning. To evaluate the development, a new apple PFT was parameterized, and the model was set up and tested using extensive site data of a mature apple orchard in northern Italy.

One-by-one parameter sensitivity analysis revealed that photosynthetic parameters and parameters associated with canopy conductance have the highest influence on GPP, NEE, LE, and yield, while phenological parameters were more influential in biomass partitioning to the different plant organs. Due to the high number of model parameters and parameter covariation, future studies could propose a more comprehensive sensitivity analysis with a training data set consisting of multiple sites, which would give more insight into model sensitivity and could further improve the parameterization.

CLM5-FruitTree was able to capture the seasonal biomass development as well as the average relative partitioning of the total biomass into the different plant organs. The inclusion of C reserves next to photosynthetic growth was imperative to enable regrowth at the end of a dormancy period and influenced LAI development, total seasonal biomass, and yield. Average simulated yield was within 2.3 % of the observation even though CLM5-FruitTree showed a lower IAV likely due to the simplification of C reserve dynamics, specific management practices, and the alternate bearing behaviour exhibited by the Fuji apple cultivar.

The new phenology and CN allocation algorithms well represented the seasonal course of carbon, water, and energy fluxes of the orchard. The magnitude of ecosystem fluxes was particularly well captured for GPP, $R_n$, LE, and ET, with correlation coefficients $>0.94$ and %bias $< \pm 5$ %. The model exhibited small biases in NEE and $R_{eco}$ that most likely were caused by the overestimation of $R_a$, especially leaf maintenance respiration, and an underestimation of $R_s$. Possible reasons for the smaller simulated contribution of $R_s$ to $R_{eco}$ could be the missing representation of the grass-covered alleys, differences in simulated and actual soil temperature or organic matter content, and oversimplification of microbial respiration processes. Additionally, large negative biases in simulated $H$ were found over most of the main irrigation season during summer as the model simulated a strong evaporative cooling of the surface temperature.

Further model developments should consider the improvement of canopy processes related to energy partitioning and the inclusion of an active ground cover in the orchard representation to improve the yearly energy budget calculations and possibly soil respiration. An explicit representation of the microbial community and a more flexible calculation of $R_a$, i.e. considering tissue age, should also be the focus of future model improvements. While the particular alternate bearing of the Fuji variety posed a challenge in this specific study, the pruning routine that is currently implemented may be sufficient for most other apple cultivars and fruit tree species for which this behaviour is less pronounced or not exhibited. However, future developments could be envisioned once the model is further tested and applied. In addition, management practices such as mowing or soil tillage could further enhance the model capability of capturing the dynamics and fate of assimilated C. Fruit thinning is another common practice in orchards, but its implementation would be more challenging, as the current model structure does not represent individual fruits. This process could however be implicitly accounted for through parameterization of the C allocation to fruits. Finally, the application of the newly developed sub-model to different geographical regions and other types of fruit trees or apple cultivars is needed to further validate the model and give more insight into the transferability of the development to different types of orchards.

Overall, our results demonstrate the ability of the newly developed CLM5-FruitTree sub-model to represent the seasonal dynamics and magnitudes of growth and ecosystem fluxes in a deciduous fruit orchard. As such, this development constitutes an important contribution to a more comprehensive representation of the agricultural land surface in CLM5 by adding a perennial, woody crop to the existing annual crop types. This will allow for a more realistic evaluation of land use and climate change effects or water availability at regional scale such as the Mediterranean or parts of China and the US, where perennial agriculture such as fruit orchards covers large parts of the agricultural landscape.

## Appendix A: Sequential model for bud break prediction

The bud break prediction in CLM5-FruitTree is based on the sequential model developed by Cesaraccio et al. (2004). Negative chill days ($C_d$) are accumulated from 1 November followed by positive anti-chill days ($C_a$) to overcome the different stages of tree dormancy, rest, and quiescence. The chilling requirement ($C_R$) defines the threshold for the accumulation of $C_d$ and is reached when $\sum C_d \leq C_R$. Thereafter, $C_a$ accumulation begins until $C_R + \sum C_a \geq 0$, at which bud break occurs. The accumulation of $C_d$ and $C_a$ on a given day is calculated from maximum ($T_x$) and minimum ($T_n$) daily air temperature as well as a temperature threshold for chill accumulation ($T_C$) and varies depending on five possible temperature cases that relate $T_x$, $T_n$, $T_C$, and $0\,°C$ to the daily mean air temperature (Table A1). The optimal values for $C_R$ and $T_C$ were calibrated based on bud break observations from 2010 to 2013 for the Adige site by minimizing the RMSE between observations and predicted bud break. The optimal value for $C_R$ was $-68$, while $T_C$ was $4\,°C$, resulting in an RMSE of $7.2\,d$.

**Table A1.** Chill day ($C_d$) and anti-chill day ($C_a$) calculation for five different temperature cases relating maximum ($T_x$) and minimum ($T_n$) air temperature to the air temperature threshold ($T_C$) and $0\,°C$; $T_M$ is the air mean temperature.

| Temperature cases | Chill days | Anti-chill days |
|---|---|---|
| $0 \le T_C \le T_n \le T_x$ | $C_d = 0$ | $C_a = T_M - T_C$ |
| $0 \le T_n \le T_C < T_x$ | $C_d = -\left[ (T_M - T_n) - \frac{(T_x - T_C)^2}{2(T_x - T_n)} \right]$ | $C_a = \frac{(T_x - T_C)^2}{2(T_x - T_n)}$ |
| $0 \le T_n \le T_x \le T_C$ | $C_d = -(T_M - T_n)$ | $C_a = 0$ |
| $T_n < 0 \le T_x \le T_C$ | $C_d = -\left[ \frac{T_x^2}{2(T_x - T_n)} \right]$ | $C_a = 0$ |
| $T_n < 0 < T_C < T_x$ | $C_d = -\frac{T_x^2}{2(T_x - T_n)} - \frac{(T_x - T_C)^2}{2(T_x - T_n)}$ | $C_a = \frac{(T_x - T_C)^2}{2(T_x - T_n)}$ |

## Appendix B: Calculation of incoming longwave radiation

Incoming longwave radiation ($LW_{in}$) can be expressed based on the Stefan–Boltzmann law as

$$LW_{in} = \varepsilon_{eff} \times \sigma \times T^4 = \varepsilon_{cs} \times F \times \sigma \times T^4, \tag{B1}$$

where $\varepsilon_{eff}$ is the effective emissivity that can be expressed by multiplying the clear-sky atmospheric emissivity $\varepsilon_{cs}$ by a cloud factor $F$ (always $\ge 1$) that expresses the increase in $LW_{in}$ under cloudy conditions, $\sigma$ is the Stefan–Boltzmann constant ($5.67 \times 10^{-8}\,W\,m^{-2}\,K^{-1}$), and $T$ is the 2 m air temperature in Kelvin.

Clear-sky emissivity was obtained using the Konzelmann et al. (1994) parameterization as follows:

$$\varepsilon_{cs} = 0.23 + 0.484 \times \left( \frac{e}{T} \right)^{\frac{1}{8}}, \tag{B2}$$

where $e$ is the vapour pressure in Pascal at 2 m.

Equation (B1) can be rearranged to obtain $F$ as follows:

$$F = \frac{LW_{in}}{\varepsilon_{cs} \times \sigma \times T^4}. \tag{B3}$$

$F$ was calculated at an hourly interval using measured $LW_{in}$ data from 2010 and $\varepsilon_{cs}$ was calculated using the above Eq. (B2).

As proposed by Sedlar and Hock (2009), in the absence of cloud data, the cloud factor $F$ can be parameterized as a function of the atmospheric transmissivity index $\tau$, which is defined as follows:

$$\tau = \frac{SW_{in}}{SW_{toa}}, \tag{B4}$$

where $SW_{in}$ is the incoming shortwave radiation, and $SW_{toa}$ is the theoretical shortwave radiation received at the top of the atmosphere.

Figure B1 shows the linear equation that was fitted to the relationship of $F$ and $\tau$ for the year 2010. For the calculation of clear-sky emissivity, all data where $\tau$ was greater than 0.7 ($N = 3863$) were considered based on the suggestion by Campbell (1985).

For the nighttime values and for very low incoming shortwave radiation ($SW_{in} < 15\,W\,m^{-2}$), $\tau$ was gap-filled with the mean of the two surrounding values to obtain a complete time series of $LW_{in}$ data. Figure B2 shows the results of the $LW_{in}$ parameterization compared to $LW_{in}$ calculated by CLM5 and to the observed data for the year 2010. As performance statistics, Pearson's $r$, the RMSE, and the percent bias are given.

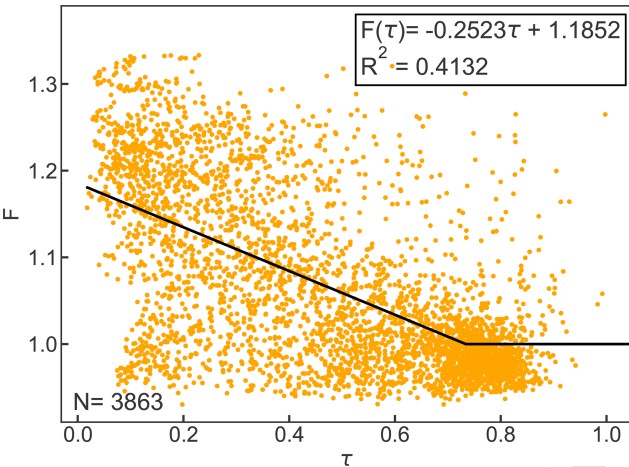

**Figure B1.** Cloud factor $F$ as a function of atmospheric emissivity $\tau$ for hourly observations. The black line represents the linear equation for $F(\tau)$ and $F \geq 1$. Clear-sky emissivity is parameterized based on Konzelmann et al. (1994).

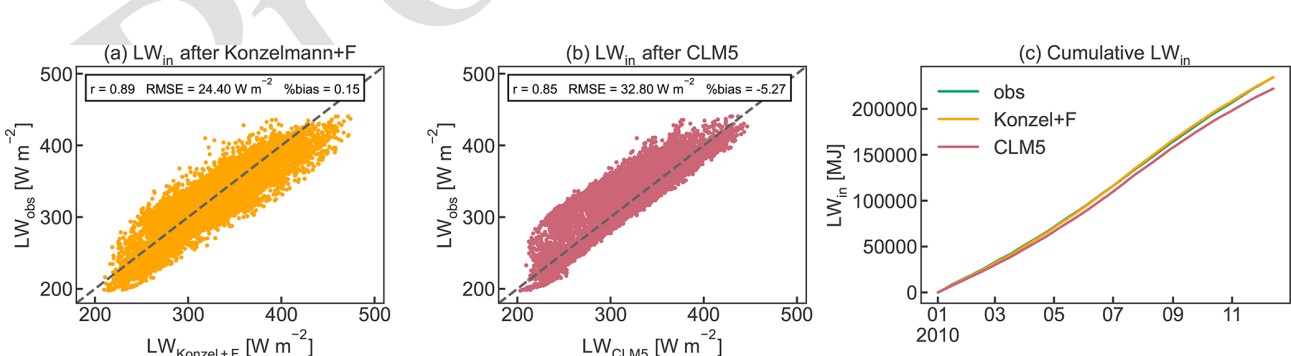

**Figure B2.** Comparison of observed $LW_{in}$ to the parameterization using **(a)** Konzelmann et al. (1994) according to Eq. (B2) and the cloud factor parameterization $F(\tau)$ and **(b)** the calculation procedure used in CLM5 as well as **(c)** cumulative observed and calculated $LW_{in}$ for 2010. Pearson's $r$, RMSE, and percent bias are given as performance statistics.

## Appendix C: Parameters used in CLM5-FruitTree and for the apple PFT

**Table C1.** Parameters adapted or added in the new CLM5-FruitTree sub-model and the apple PFT, including phenology, CN allocation, photosynthesis, vegetation structure, as well as optical and respiration parameters. Parameters were adjusted based on field observations or literature values and are listed with their definition, unit, value, and references to the literature.

| Parameter | Definition | Unit | Value | Reference |
|---|---|---|---|---|
| Phenological parameters | | | | |
| baset | Base temperature for GDD accumulation | °C | 4 | Based on commonly used values for apple trees (Reyes et al., 2016; Díez-Palet et al., 2019; Penzel et al., 2020) |
| crequ | Chilling requirements for bud break of fruit tree crops | Unitless | −68 | Calibrated using bud break dates from Zanotelli et al. (2013, 2015) and the sequential model (Cesaraccio et al., 2004) |
| crit_temp | Critical temperature to initiate leaf senescence for fruit tree crops | K | 278.15 | Adjusted based on LAI measurements (Zanotelli et al., 2013) |
| grnfill* (GDDfruit) | GDD needed from bud break to beginning of fruit development | ° days | 400 | Based on observed and commonly used values for apple trees (Zanotelli et al., 2013; Lakso et al., 2000; Neumann, 2020; Penzel et al., 2020) |
| grnrp* (GDDripe) | GDD needed from bud break to the fruit-ripening phase | ° days | 1100 | Based on observed and commonly used values for apple trees (Lakso et al., 2000; Zanotelli et al., 2013; Neumann, 2020; Penzel et al., 2020) |
| huileaf (GDDleaf) | GDD accumulated at the moment of bud break (end of dormancy period) | ° days | – | Calculated based on the sequential model for bud break prediction (Cesaraccio et al., 2004) |
| hybgdd* (GDDmat) | GDD needed from bud break until fruit harvest | ° days | 2880 | Based on observed and commonly used values for apple trees (Lakso et al., 2000; Zanotelli et al., 2013; Neumann, 2020; Penzel et al., 2020) |
| laimx* | Maximum leaf area index | $m^2\,m^{-2}$ | 3 | Based on observed and commonly used values for apple trees (Valancogne et al., 1999; Li et al., 2002; Zanotelli et al., 2013) |
| lfmat* (GDDlfmat) | GDD needed from bud break to canopy maturity | ° days | 1350 | Based on observed and commonly used values for apple trees (Lakso et al., 2000; Zanotelli et al., 2013; Neumann, 2020; Penzel et al., 2020) |
| max_NH_harvest_date | Maximum harvest date for the Northern Hemisphere (NH) | Date (md) | 1015 | Based on typical harvest dates in the NH |
| max_NH_planting_date | Maximum planting date for the NH | Date (md) | 101 | Only needed for orchard establishment and initiation of the sequential model for bud break; tree is still dormant |
| min_NH_planting_date | Minimum planting date for the NH | Date (md)[TS3] | 101 | Only needed for orchard establishment and initiation of the sequential model for bud break; tree is still dormant |
| mxmat | Maximum orchard age | d | 9125 | Based on common values for apple orchards (Lakso et al., 2000; Zanotelli et al., 2013; Penzel et al., 2020) |

| Parameter | Definition | Unit | Value | Reference |
|---|---|---|---|---|
| ndays_stor | Length of period for storage growth of fruit tree crops | d | 50 | Based on common values for fruit orchards (Kozlowski, 1992; DeJong and Grossman, 1994; Wünsche and Lakso, 2000) |
| perennial | Binary flag for perennial crop phenology | Unitless | 1 | |
| root_dmx | Maximum rooting depth of crops | m | 0.8 | Based on observed rooting depth (Zanotelli et al., 2013) |
| rootprof_beta* | Rooting beta parameter for C and N vertical discretization | Unitless | 0.964 | Calibrated based on root sampling campaign of root mass up to 60 cm (Zanotelli, 2010, unpublished data) |
| woody | Binary flag for woody life form | Unitless | 1 | |
| C and N allocation parameters | | | | |
| aleaff* | Final leaf allocation coefficient | Unitless | 0.01 | Adjusted based on monthly biomass measurements (Zanotelli et al., 2013) |
| aleafstor* | Leaf allocation coefficient for storage post-harvest used in CN allocation | Unitless | 0.3 | Adjusted based on monthly biomass measurements (Zanotelli et al., 2013) |
| allconss* | Power to control the shape of the stem allocation curve | Unitless | 1.5 | Adjusted based on monthly biomass measurements (Zanotelli et al., 2013) |
| arootf* | Root allocation coefficient at start of fruit development | Unitless | 0.2 | Adjusted based on monthly biomass measurements (Zanotelli et al., 2013) |
| arootf2* | Final root allocation coefficient until harvest | Unitless | 0.08 | Adjusted based on monthly biomass measurements (Zanotelli et al., 2013) |
| arooti* | Initial root allocation coefficient | Unitless | 0.7 | Adjusted based on monthly biomass measurements (Zanotelli et al., 2013) |
| astemf* | Final stem allocation coefficient | Unitless | 0.22 | Adjusted based on monthly biomass measurements (Zanotelli et al., 2013) |
| bfact* | Exponential factor used for fraction allocated to leaf | Unitless | $-0.5$ | Adjusted based on monthly biomass measurements (Zanotelli et al., 2013) |
| declfact* | Decline factor to control the shape of the stem allocation curve | Unitless | 4 | Adjusted based on monthly biomass measurements (Zanotelli et al., 2013) |
| fcur* | Fraction of C and N allocated to the displayed pools | Unitless | 0.95 | Tuned based on observed LAI and yield data (Zanotelli et al., 2013) |
| fleafi* | Initial leaf allocation coefficient | Unitless | 0.85 | Adjusted based on monthly biomass measurements (Zanotelli et al., 2013) |
| flivewd | Fraction of new wood that is live | | 0.15 | Same as BDT in CLM5 |
| frootCN | Fine root C : N ratio | $gC\,gN^{-1}$ TS4 | 32 | Average of six measurements (Zanotelli, 2010, unpublished data) |
| grainCN* | Fruit C : N ratio | $gC\,gN^{-1}$ | 139 | Average of six measurements (Zanotelli, 2010, unpublished data) |
| leafCN* | Leaf C : N ratio | $gC\,gN^{-1}$ | 19.7 | Average of six measurements (Zanotelli, 2010, unpublished data) |
| lflitCN | Litter C : N ratio | $gC\,gN^{-1}$ | 60 | Average of four measurements (Zanotelli, 2010, unpublished data) |
| livewdCN | Livewood C : N ratio | $gC\,gN^{-1}$ | 60 | Average of six measurements (Zanotelli, 2010, unpublished data) |
| transplant | Initial carbon for crops transplanted from nursery | gC TS5 | 5 | |
| Photosynthetic parameters | | | | |
| i_vcad* | Intercept of the relationship between leaf N per unit area and Vcmax25top | $\mu molCO_2\,m^{-2}\,s^{-1}$ | 5.2 | Adjusted in between BDT and crop |
| medlynslope* | Medlyn slope of conductance–photosynthesis relationship | $\mu molH_2O\,\mu molCO_2^{-1}$ | 8.2 | Tuned based on observed GPP and ET data (Zanotelli et al., 2015) |
| s_vcad* | Slope of the relationship between leaf N per unit area and Vcmax25top | $\mu molCO_2\,s^{-1}\,gN^{-1}$ | 34 | Tuned based on observed LAI and yield data (Zanotelli et al., 2013) |
| slatop* | Specific leaf area at top of canopy | $m^2\,gC^{-1}$ | 0.028 | Mean value for the growing season based on LAI and leaf biomass measurements (Zanotelli et al., 2013) |

| Parameter | Definition | Unit | Value | Reference |
|---|---|---|---|---|
| Vegetation structure and management | | | | |
| displar | Ratio of displacement height to canopy top height | Unitless | 0.67 | Same as BDT in CLM5 |
| mulch_pruning | Binary flag for mulching (1) or export (0) of pruning material | Unitless | 1 | Based on reported organic farming practices (Zanotelli et al., 2013) |
| prune_fr | Fraction of dead stem that is pruned | Unitless | 0.85 | Based on reported pruning quantity (Zanotelli et al., 2015) |
| nstem | Planting density | $\# \, m^{-2}$ | 0.33 | Based on reported planting density (Zanotelli et al., 2013) |
| taper | Ratio of stem height to radius at breast height | TS6 | 120 | Based on reported tree allometry and height (Zanotelli et al., 2013) |
| xl* | Leaf/stem orientation index | Unitless | 0.25 | Same as BDT in CLM5 |
| z0mr* | Ratio of momentum roughness length to canopy top height | Unitless | 0.06 | Based on average values reported for apple (de la Fuente-Sáiz et al., 2017) and citrus (Tanny and Cohen, 2003) orchards |
| ztopmx | Maximum canopy height for crops | m | 3.6 | Based on reported tree heights (Zanotelli et al., 2013) |
| Optical parameters | | | | |
| rholnir* | Leaf reflectance: near IR | Fraction | 0.5 | Based on average values for apple trees (Bastías and Corelli-Grappadelli, 2012) |
| rholvis* | Leaf reflectance: visible | Fraction | 0.1 | Based on average values for apple trees (Bastías and Corelli-Grappadelli, 2012) |
| rhosnir* | Stem reflectance: near IR | Fraction | 0.39 | Same as BDT in CLM5 |
| rhosvis* | Stem reflectance: visible | Fraction | 0.16 | Same as BDT in CLM5 |
| taulnir* | Leaf transmittance: near IR | Fraction | 0.3 | Based on average values for apple trees (Bastías and Corelli-Grappadelli, 2012) |
| taulvis* | Leaf transmittance: visible | Fraction | 0.04 | Based on average values for apple trees (Bastías and Corelli-Grappadelli, 2012) |
| tausnir* | Stem transmittance: near IR | Fraction | 0.001 | Same as BDT in CLM5 |
| tausvis* | Stem transmittance: visible | Fraction | 0.001 | Same as BDT in CLM5 |
| Respiration | | | | |
| FUN_fracfixers* | The maximum fraction of assimilated carbon that can be used to pay for N fixation | Fraction | 0.25 | Same as BDT in CLM5 |
| lmr_intercept_atkin | Intercept in the calculation of the top of canopy leaf maintenance respiration base rate | $\mu mol \, CO_2 \, m^{-2} \, s^{-1}$ | 1.756 | Same as BDT in CLM5 |

* Parameters included in the sensitivity analysis.

*Code availability.* The new CLM5-FruitTree sub-model is freely available via Zenodo at https://doi.org/10.5281/zenodo.6595378 TS7 (Dombrowski, 2022).

*Data availability.* Data from the Laimburg weather station were kindly provided by the station operator Martin Thalheimer (Research Centre for Agriculture and Forestry, Laimburg, Bolzano) upon request. All other data from the apple orchard in South Tyrol, Italy were provided by Damiano Zanotelli and his team and are licensed under the Creative Commons Attribution 4.0 International License (https://creativecommons.org/licenses/by/4.0/, last access: TS8). They can be made available upon request.

*Author contributions.* OD developed and modified the code for the sub-model, designed, performed, and analysed the simulations, and prepared the original draft of the manuscript. HB, HJHF, and CB supervised the research, and, together with DZ, contributed to the manuscript writing through review and editing.

*Competing interests.* The contact author has declared that neither they nor their co-authors have any competing interests.

*Acknowledgements.* The authors are grateful to Damiano Zanotelli and his team for providing the field data of the apple orchard in South Tyrol, Italy. The authors thank Yuanchao Fan for sharing the source files for his development of CLM-Palm (Fan et al., 2015) that aided the development of the new CLM5-FruitTree sub-model. Furthermore, the authors thank Martin Thalheimer for providing the meteorological data from Laimburg meteorological station. TS9

*Financial support.* This research has been supported by Horizon 2020 (ATLAS (grant no. 857125)) and by the Deutsche Forschungsgemeinschaft under Germany's Excellence Strategy (grant no. EXC-2070-390732324-PhenoRob) TS10.

The article processing charges for this open-access publication were covered by the Forschungszentrum Jülich. TS11

*Review statement.* This paper was edited by Christoph Müller and reviewed by two anonymous referees.

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

## Remarks from the typesetter

**TS1** Please note that vector graphics (*.eps, *.ps) cannot be included in the PDFLaTeX for technical reasons. In addition, *.pdf figures cannot be included in the PDFLaTeX since certain fonts or other content cannot be embedded and such content would then not show up in some browsers or *.pdf viewers. As a result, affected figures might appear incomplete to some readers. Therefore, we only include *.png and *.jpg figures in the article *.pdf. However, since we also publish all articles in full-text HTML, we will provide your vector graphics as high-resolution figures so that readers are able to download and enlarge the figures for re-use (see e.g. https://gmd.copernicus.org/articles/15/3879/2022/). Please note that this high-resolution download is only possible if your figure has the Creative Commons Attribution 4.0 License (CC BY) applied. This is the case for the figures compiled by you or your co-authors. If you cite a figure from another paper that is not distributed under the Creative Commons Attribution License, the figure is identified as protected and the download link will be hidden.