# Peer review of "CLM5-FruitTree: A new sub-model for deciduous fruit trees in the Community Land Model (CLM5)"

_Geoscientific Model Development, 2022_

## Referee Comment (RC1)

**Review of manuscript for GMD:**

"CLM-FruitTree: A new sub-model for deciduous fruit trees in the Community Land Model (CLM5)" (Dombrowski et al.)

This manuscript describes the development of a fruit-tree sub-model as part of CLM5, a well-established land and vegetation model. As pointed out by the authors, the inclusion of new agricultural vegetation types in large scale simulation models is an important advancement for understanding and quantifying their role in many biophysical earth-system processes as well as improve the representation of the agricultural sector production. Overall, the manuscript is of good scientific quality.

One limitation of the study is that it is performed on a single point for only a few years, which limits the possibility to evaluate its validity under different conditions. On the other hand, an extremely rich dataset of measurements is used to calibrate and validate the new model. This gives confidence on the representation of processes, such as GPP, NPP, Carbon allocation and crop yields.

The results are well presented with a good structure and informative figures. Yet, some aspects have not been covered, hampering full understanding of the conceptual model and the reproducibility of results:

1) One of the greatest challenges in modelling orchards is the representation of the canopy structure, which is not closed and uniform in space, as usually assumed for arable crops and for natural forest. This is a crucial aspect that affects the way radiation is intercepted by the crop canopy. In the paper, it is not described what assumptions have been made regarding radiation interception and whether changes to the CLM5 model have been necessary.

2) It is not mentioned what are the structural characteristics accounted for to represent the orchard in the model. Particularly:
   - The planting density
   - The in-row and between-rows planting distances and the ground covered by the canopy
   - How large are seedlings transplanted from the nursery and what is their allometry at establishment (e.g. tree height, stem diameter, LAI, sapwood/heartwood partition, ....)?

3) An important process of fruit-tree species is flowering, which is not explicitly represented in the CLM-FruitTree model. Although, it is clearly an acceptable simplification in this kind of model, the assumptions behind this choice (e.g. optimal pollination, compensation effects between fruit numbers and size, ... ) and its implication should be presented and discussed.

Below I provide further comments on specific sections, lines and figures that need improvement.

**Methods**

**Structure**

In general section 2.1 gives the motivation for developing the CLM-FruitTree, but I think it would be better structured in this form: "to simulate fruit trees we need a model that does XYZ; CLM5 with its improvement is a good base for this, indeed it includes ABC; yet, it still misses ZYX that we implement in this paper." Otherwise it is not clear why you describe those aspects of CLM5.

Section 2.2 is not very informative in terms of model conceptualization. Please, use this section to (1) give an overall description of what system your model describes (e.g. what kind of apple orchard, extensive / intensive), (2) explain which components of the system should the model represent well (e.g. it should be at least good at simulating average yields and carbon stocks), (3) describe the model

concept, preferably referring to the diagram displayed in Fig. 1. Here it would be a good place also to define the three C pools that are mentioned also in 2.2.2 without a proper explanation.

Section 2.2.1: It is good to start off with phenology. Please, stick to that and do not mix phenology with growth processes. E.g. why is initial biomass mentioned in L157? Similar for L163. Maybe, put these into a paragraph at the end of 2.2.1, describing growth processes triggered by phenological events.

Section 2.2.3: Please, restructure the paragraph L219-232 to make clear what is common practice in the "real world" and what is implemented in the model. First explain the common practice and then what's in the model.

**Line-specific comments**

L97-99: As the names of these pools appear here for the first time, the sentence is a bit confusing. I would suggest to clarify the sentence as follows, use italic for the pool names and refer to later sections for additional details: "Once a new onset growth period is initiated, C and the corresponding N fluxes occur out of a *storage* pool, which are temporarily stored into an intermediate pool (*transfer* pool) and then gradually transferred to the *display* growth pools (see section XYZ for details)."

L100: Are there other stoichiometric relationships other then C:N ratios? If yes, the sentence is fine, otherwise please, remove stoichiometric relationships.

L101-102: Sounds like a repetition of L93, please merge the two.

L122: Unclear whether the management options are related to phenology management (e.g. choice of cultivar?) or to other management practices somewhat connected to phenology (e.g. pruning?).

L123-124: "were modified" is too vague. As you don't have space here to go into details, I'd suggest to be brief but explanatory, e.g. "CN fluxes and allocation were modified to fit ....".

L124-128: These are very technical details and not so much part of the model conceptualization. I wonder whether it would be possible to make a separate section on "technical implementation" to describe these.

L135-136: This seems quite long for modern orchards. What kind of orchards are you simulating? Intensive / extensive, low / high density, what are the assumptions on the rootstock?

L158: Apple growth or apple-tree growth?

L163: how large is the portion of C transferred?

L164: Please, provide a reference or justification for the 50 days assumption.

L165-167: From this description ("fruit starts 4-5 w after bud break", "leaf senescence occurs after harvest") it does not seem that leaves and fruits development are independent from each other.

L186: Shouldn't "except for fruits where all allocated C is assigned to the displayed pool" be part of the previous sentence?

L199-200: Allocation to fine roots and stem decline, not the root and stem pool themselves, right?

L210: Please, expand a bit on the N retranslocation strategy, not just by referring to Lawrence et al., 2018. Doesn't this belong to 2.2.2 as it refers to N allocation. Then you could call section 2.2.3 simply "Representation of management practices" and include here details of all managements, including the assumed orchard design (planting densities, raw arrangement, training system).

L220: What do you mean by "dead stem"? Usually pruning is meant to remove living branches. Might be that CLM does not explicitly distinguish stem and later branches. Yet, more explanations are needed here to justify the implemented pruning routine.

L313: for clarity, X and deltaX also need to be defined.

**Results and Discussion**

**Figures**

Fig. 3: To improve readability, I suggest to name the parameters with their extended names and the short name in parenthesis, e.g. gross primary production (GPP), directly in the plot and not in the caption.

Fig. 5: It is not clear whether the x-axis ticks refer to the beginning/midday/end of the months. Moreover, more ticks would help reading the timing of events, e.g. when is full canopy development reached.

Fig. 6: According to Zanotelli et al., 2019 (section 2.1), yields in 2015 has been 63 t ha-1. Please, double check.

**Line-specific comments**

L368-370: why "primarily". Isn't it all allocated to those organs? In the methods it is stated that storage Carbon is used for growth of all organs except fruits in the first 50 days after bud breaks. Moreover, from Fig.4 it looks like growth is supported by storages way beyond early May, rather until early June. When the fruit curve is already taking off.

L372: In Fig. 4, leaf biomass seems to reach the plateau earlier, in June. The peak in July better refers to observations, correct?

L390: for clarity, replace "light pruning" with "a lighter pruning compared to the previous year" or similar. Moreover, if such lighter pruning happens on-field every second year, it should not sound like it was an extraordinary event in 2011 that cannot be captured by the model, but rather a flexibility in management that is not well represented in the model. If the model with fixed management "sees" an alternation of "good" and "bad" years, it could mean that it represents processes well, and it has a too simplified management that leaves room for improvement.

L407-409: Not clear. Usually management should aim at reducing yield variability for both arable and perennial crops, e.g. irrigation to reduce precipitation variability, pruning to reduce alternate bearing of fruit trees, etc.

L438: what is indicated in parenthesis? Standard deviation, range, ...

L457-463: This paragraph is unclear and hard to follow. Please, report measured values along with observed values and vice versa. E.g. in L457, how much is Rs and its share in Reco for the simulations? Please, move "In contrast, simulated Reco for the same year [...]" of L459 right after "[...] measurements within the orchard (total soil respiration)." in L458.

L472: The representation of the different components of respiration in CLM should be explained in the methods, as this is one of the metrics to evaluate the new model implementation.

L462 & L478: It is not clear why citrus orchards should be a valid reference also for apple orchards. The discussion needs to be improved, bringing more references (e.g. on more tree species) if existing or justifying why citrus trees can be a good reference.

L536: In the figure soil moisture (SM) is called soil water content (SWC). Please, be consisent.

---

## Referee Comment (RC2)

**Review: CLM-FruitTree manuscript (gmd-2022-41)**

**General comments**

In this manuscript, Dombrowski et al. introduce a new kind of crop to the Community Land Model (CLM): fruit trees. They present a parameterization for apple trees, but note that the code they've written could be applied to other fruit-bearing trees. This represents an important step forward for CLM, which, like many global gridded crop models, has heretofore mostly excluded anything woody or perennial. Incorporating this development into CLM, especially with additional types of fruit trees, would enable the simulation of crops important not just for food security in terms of calories, but also in terms of nutrition and economic productivity.

The model performs well compared to observations in terms of most evaluated metrics, especially yield. The authors do a good job in most cases of identifying discrepancies and suggesting hypotheses for their causes, which are often structural issues with CLM which it would be outside the scope of this work to resolve. The manuscript does unfortunately use just one real-life orchard for parameterization and evaluation of the model; fully incorporating apples as a scientifically-supported crop within CLM will likely take more effort to generalize the parameterization. But the work presented here represents a significant enough advance that it does merit publication in *GMD*. Importantly, the authors performed and presented the results of a basic sensitivity analysis, which will aid in future parameterization work.

The manuscript is laid out logically, well-written, and well-supported by the provided figures. Most of my suggestions are relatively minor, and thus I recommend this manuscript be ***published pending minor revisions***.

**Specific comments**

My only really substantive comments have to do with the exploration of discrepancies between the simulation and observations:

- L390-395: The simulated LAI in 2011 is too low, which the authors suggest could be due to pruning having been performed in the real world. But is the "alternate bearing behavior" something the authors actually expected the model to represent? If so, how? It seems like something that would need to be explicitly coded in.
- L405-425: I would think that real-world management practices such as fruit thinning have the aim of *reducing* interannual variability (IAV), but it sounds like the authors are suggesting that CLM's IAV is too low because they're *not* represented. In general, it seems like missing physiological processes and/or extreme event representation would be more to blame for too-low IAV.
- L472–480: It's unclear from the data presented here that autotrophic respiration actually is too high in CLM5. Yes, it's too high a proportion of total ecosystem respiration, but the authors have established that soil respiration is too low. This paragraph should discuss absolute units in addition to relative ones.

In addition, some general comments:

- Please consider making your parameterization script(s) available as well.
- According to *GMD* rules, the title needs a version number for CLM-FruitTree. Ideally this would correspond to a release tag in the GitHub repository.

**Technical corrections and minor comments**

- L17: EC is undefined
- L33: Apostrophe should be a comma
- L57: Adding abbreviation of "(LPJmL)" might be useful
- L67: "buildup" would be clearer than "deposition"
- L92 and throughout: Should also cite Lombardozzi et al. (2020, *JGR: Biogeosci*: "Simulating Agriculture in the Community Land Model Version 5"), in addition to/instead of Lawrence et al. (2018)
- L116-7: "active growth in the current season" is unclear
- L144: "full bloom" is unclear
- L150 (Fig. 1):
    - "brown" would be more accessible than "ochre" for non-native English readers
    - "DISPLAY" is unclear. Is this a standard CLM term? If so, define it; if not, another word would be better.
    - Unclear from this that each plant part has its own storage and transfer pool (except, presumably, fruits)
- L162-5: Is *all* the C in storage pools transferred over the 50 days? If not, what "portion" is?
- L169-70: Are these GDD parameters something that can be set for each fruit tree PFT, or are hard-coded?
- L173: "offset"? Is this the same as senescence?
- L180 (Fig. 2)
    - What are the bars, exactly? Period of growth?
    - Would be clearer and more consistent for "canopy development" to just be "leaves"
- L211-2: "CLM-FruitTree adopts the same N retranslocation strategy as used in the BDT phenology," but above (L149) it says "minor adaptations" were made.
- L228: "effects" should be "affects"
- L270: Was the forcing de-trended during spinup?
- L393-5: "In consequence to" should be "Due to" or "As a consequence of".
- L399-400: This sentence is unclear.
- L405: Delete "at".
- L437: "Returns **to** positive"
- L520: "phenomena" should be "phenomenon".
- L531-2: This sentence is unclear. "Patchy" what?
- L579-580: But also overestimation of soil respiration!
- L588-9: What about pruning and fruit thinning?

- L630 (Fig. B1): Please use a thicker font for this (or maybe a higher-res image); it disappears at medium zoom levels.

Figure B1 shows the linear equation that was fitted to the relationship of F and τ for the year 2010. For the calculation of clear-sky emissivity, all data where τ was greater than 0.7 (N=3863) was considered based on the suggestion by Campbell (1985).

[Figure]

- L635 (Fig. B2): Same issue as Fig. B1.

---

## Author Comment (AC1)

**Answer to Reviewer #1 of manuscript for GMD:**

"CLM-FruitTree: A new sub-model for deciduous fruit trees in the Community Land Model (CLM5)" (Dombrowski et al.)

➢ We thank the reviewer for taking the time to read the manuscript and for providing constructive feedback on our work. In the following we are presenting our preliminary responses to the reviewer's comments. The revision and resubmission of the paper will follow once we received the reviews of the other reviewers. We hope the reviewer will find the comments and concerns addressed appropriately in the meantime.

This manuscript describes the development of a fruit-tree sub-model as part of CLM5, a well-established land and vegetation model. As pointed out by the authors, the inclusion of new agricultural vegetation types in large scale simulation models is an important advancement for understanding and quantifying their role in many biophysical earth-system processes as well as improve the representation of the agricultural sector production. Overall, the manuscript is of good scientific quality.
One limitation of the study is that it is performed on a single point for only a few years, which limits the possibility to evaluate its validity under different conditions. On the other hand, an extremely rich dataset of measurements is used to calibrate and validate the new model. This gives confidence on the representation of processes, such as GPP, NPP, Carbon allocation and crop yields.

➢ We are pleased that the reviewer recognises the quality of our scientific work and the uniqueness of the dataset in terms of range and detail of the available measurements, which has provided a unique opportunity to develop the new CLM5 sub-model CLM-FruitTree. This work focuses on the sub-model description, but we agree that the validity of CLM-FruitTree should be further tested with datasets from other geographic regions, longer time series and different orchard types. A major challenge we see here is that data sets with similar detail as the one used in this study are hardly available for orchard ecosystems at this point. Therefore, further validation and testing of CLM-FruitTree was beyond the scope of this paper and should be rather accomplished by future studies. Nonetheless, we will further stress this point in the revised manuscript.

The results are well presented with a good structure and informative figures. Yet, some aspects have not been covered, hampering full understanding of the conceptual model and the reproducibility of results:

➢ We will address the specific comments of the reviewer #1 one by one in the following.

1) One of the greatest challenges in modelling orchards is the representation of the canopy structure, which is not closed and uniform in space, as usually assumed for arable crops and for natural forest. This is a crucial aspect that affects the way radiation is intercepted by the crop canopy. In the paper, it is not described what assumptions have been made regarding radiation interception and whether changes to the CLM5 model have been necessary.

➢ Canopy structure is indeed a crucial aspect of modelling radiation interception and energy partitioning within the crop canopy. CLM5 currently is still limited to the assumption of a closed canopy structure that is uniform in space. While future developments towards integrating multi-layer schemes for canopy processes and explicit representation of the canopy to improve related processes are desirable, they could not be realized in this development, and we did not make any changes to the existing calculations of momentum, heat, and water fluxes. We adapted the *z0mr* parameter (the ratio of momentum roughness length to canopy top height) and *displar* (the ratio

of displacement height to canopy top height) to account for differences to arable crops and natural forests but we also acknowledge the limitations of not representing the orchard structure more realistically in L531-534 and L585 of the manuscript. Nonetheless, we will add a sentence in the section model conceptualization to more clearly state the assumptions that were made.

2) It is not mentioned what are the structural characteristics accounted for to represent the orchard in the model. Particularly:

➢ This comment relates to comment 1) above and is thus partially explained in the answer above. In addition, we will provide the following information on structural features in the revision:

• The planting density

➢ Planting density is specified via the *stocking* parameter used also for natural forests in CLM5*.* This parameter is currently not part of the parameter file but we will add it to the parameter file for the revised version of the code and manuscript. Currently the *stocking* parameter together with the *taper* parameter (ratio of stem height to radius) present the very simplified allometry of trees in CLM5 and are only used for the calculation of top canopy height.

• The in-row and between-rows planting distances and the ground covered by the canopy

➢ There are no parameters defining row spacings instead the area covered by the plants is defined by leaf and stem area index in a uniform way.

• How large are seedlings transplanted from the nursery and what is their allometry at establishment (e.g. tree height, stem diameter, LAI, sapwood/heartwood partition, ....)?

➢ Tree allometry is calculated based on the parameter values used for *stocking*, *taper,* and *transplant* (defines initial leaf and dead stem biomass) resulting in an initial tree height of around 40 cm, a stem diameter of 7 mm and LAI of 0 since trees are transplanted during winter. Seedling allometry has little effect on the biomass growth and yield of the adult trees as in any case trees reach their maximum canopy height and full LAI within the first couple of years after transplanting. We assume that for most applications of the presented developments, seedling size will be of minor interest and instead the focus will be on yield, biomass growth, C turnover etc.

3) An important process of fruit-tree species is flowering, which is not explicitly represented in the CLM-FruitTree model. Although, it is clearly an acceptable simplification in this kind of model, the assumptions behind this choice (e.g. optimal pollination, compensation effects between fruit numbers and size, ... ) and its implication should be presented and discussed.

➢ As recognized by the reviewer, the explicit representation of flowering is out of the scope for a development within a land surface model aiming at large scale simulations and processes at ecosystem level. Consequently, CLM-FruitTree does not produce information on fruit size or number but only on total yield. Simulations should be calibrated against observed yield by adjusting the phenological and CN allocation parameters related to fruit growth. Effects of non-optimal pollination or fruit drop later in the season are hence not captured by the model development, which could result in lower simulated inter-annual yield variability. Following the reviewer's suggestion, we will briefly present and discuss the decision to not explicitly represent flowering in the manuscript.

Below I provide further comments on specific sections, lines and figures that need improvement.

**Methods**

**Structure**

In general section 2.1 gives the motivation for developing the CLM-FruitTree, but I think it would be better structured in this form: "to simulate fruit trees we need a model that does XYZ; CLM5 with its improvement is a good base for this, indeed it includes ABC; yet, it still misses ZYX that we implement in this paper." Otherwise it is not clear why you describe those aspects of CLM5.

➢ We thank the reviewer for the suggestion. We will edit this section to make clear why we introduce the different vegetation types described in CLM5. We will then explain which aspects of them and to what extent they can be used to model fruit trees before stressing the limitations of the existing vegetation characterizations.

Section 2.2 is not very informative in terms of model conceptualization. Please, use this section to (1) give an overall description of what system your model describes (e.g. what kind of apple orchard, extensive / intensive), (2) explain which components of the system should the model represent well (e.g. it should be at least good at simulating average yields and carbon stocks), (3) describe the model concept, preferably referring to the diagram displayed in Fig. 1. Here it would be a good place also to define the three C pools that are mentioned also in 2.2.2 without a proper explanation.

➢ Thank you for this comment, we agree that this section would benefit from adding more details on the model concepts. We will provide additional information as suggested. As the three C pools are first mentioned in section 2.1, we will define them there.

Section 2.2.1: It is good to start off with phenology. Please, stick to that and do not mix phenology with growth processes. E.g. why is initial biomass mentioned in L157? Similar for L163. Maybe, put these into a paragraph at the end of 2.2.1, describing growth processes triggered by phenological events.

➢ Sentences in L157 and L163 will be rephrased and moved to the first paragraph of 2.2.2.

Section 2.2.3: Please, restructure the paragraph L219-232 to make clear what is common practice in the "real world" and what is implemented in the model. First explain the common practice and then what's in the model.

➢ We will restructure this paragraph following the suggestion of the reviewer.

**Line-specific comments**

L97-99: As the names of these pools appear here for the first time, the sentence is a bit confusing. I would suggest to clarify the sentence as follows, use italic for the pool names and refer to later sections for additional details: "Once a new onset growth period is initiated, C and the corresponding N fluxes occur out of a *storage* pool, which are temporarily stored into an intermediate pool (*transfer* pool) and then gradually transferred to the *display* growth pools (see section XYZ for details)."

➢ We will address this comment by defining the different carbon pools in section 2.1.

L100: Are there other stoichiometric relationships other then C:N ratios? If yes, the sentence is fine, otherwise please, remove stoichiometric relationships.

➢ The sentence will be modified as follows: "During the active growth period, C and corresponding N storage pools of the individual plant organs are replenished based on specified C:N ratios of each plant organ."

L101-102: Sounds like a repetition of L93, please merge the two.

➢ We believe the reviewer is referring to the sentence in L95-96. We will shorten the sentence by removing "between different plant and litter pools." to avoid any repetition.

L122: Unclear whether the management options are related to phenology management (e.g. choice of cultivar?) or to other management practices somewhat connected to phenology (e.g. pruning?).

➢ For the sake of greater clarity, we will amend the sentence as follows: "we introduced a new phenology subroutine including triggers for seasonal orchard management practices"

L123-124: "were modified" is too vague. As you don't have space here to go into details, I'd suggest to be brief but explanatory, e.g. "CN fluxes and allocation were modified to fit ....".

➢ The sentence will be edited as follows: "the CN allocation module as well as associated modules (including C and N state and flux updates, vegetation structure, and respiration) were modified to reproduce the growth dynamics of fruit trees."

L124-128: These are very technical details and not so much part of the model conceptualization. I wonder whether it would be possible to make a separate section on "technical implementation" to describe these.

➢ We will rename the paragraph "Model conceptualization and technical implementation". In this way we will discuss the used modelling concepts and at the same time briefly explain their technical implementation where necessary to avoid repetition and a separate paragraph.

L135-136: This seems quite long for modern orchards. What kind of orchards are you simulating? Intensive / extensive, low / high density, what are the assumptions on the rootstock?

➢ The lifespan can be adjusted to any value as desired by the user. The given years are indeed on the high end for organic or semi-extensive system while intensive orchards typically have shorter lifespans of around 15 years. To avoid confusion, we will modify the sentence to "Once planted, the orchard remains productive according to a user-defined lifespan depending on production system, typically 15-20 years for intensive systems and up to 30 years for extensive systems" and will provide a reference. There are no specific assumptions made on the rootstock but the effect of different root stocks in terms of tree height and rooting depth can be set by the user via the respective parameters (*ztopmx* and *root_dmx*)

L158: Apple growth or apple-tree growth?

➢ Tree growth is meant, we will correct this.

L163: how large is the portion of C transferred?

➢ The transferred portion is 0.5 based on the assumption that resources are partially mobilized to support growth of the new season but lacking more specific knowledge on that fraction. This is the default fraction used by CLM5 in the seasonal deciduous trees algorithm.

L164: Please, provide a reference or justification for the 50 days assumption.

➢ This parameter was calibrated based on the biomass measurements and the estimate by Zanotelli et al. 2013 that apple trees use stored carbohydrates in the first two months after budburst. We will add this information.

L165-167: From this description ("fruit starts 4-5 w after bud break", "leaf senescence occurs after harvest") it does not seem that leaves and fruits development are independent from each other.

➢ Independent here meaning that they can evolve in parallel as opposed to the standard crop module were grain fill starts once leaf area development is finished. We will reformulate this to avoid confusion.

L186: Shouldn't "except for fruits where all allocated C is assigned to the displayed pool" be part of the previous sentence?

➢ We will change the sentence as follows: "The remainder is allocated to the displayed C pools while for fruits all allocated C is assigned to the displayed pool. "

L199-200: Allocation to fine roots and stem decline, not the root and stem pool themselves, right?

➢ Yes indeed, we will insert "allocation" in the sentence.

L210: Please, expand a bit on the N retranslocation strategy, not just by referring to Lawrence et al., 2018. Doesn't this belong to 2.2.2 as it refers to N allocation. Then you could call section 2.2.3 simply "Representation of management practices" and include here details of all managements, including the assumed orchard design (planting densities, raw arrangement, training system).

➢ We will move N retranslocation to section 2.2.2 as suggested by the reviewer and add some more detail: "The N retranslocation algorithm removes N from the falling litter based on leaf and litter CN ratios and the available C to pay for the extraction of N from increasingly more recalcitrant litter pools."

L220: What do you mean by "dead stem"? Usually pruning is meant to remove living branches. Might be that CLM does not explicitly distinguish stem and later branches. Yet, more explanations are needed here to justify the implemented pruning routine.

➢ CLM5 converts all live stem biomass to dead stem biomass at the end of a growing season to account for reduced maintenance cost of different tissue ages. In fact, no maintenance respiration is assumed for dead woody tissue. As such, the effect of the pruning routine is solely on the carbon pools while not affecting maintenance costs of the trees. We will add an explanation of this in section 2.2.2 upon first mention of the dead stem.

L313: for clarity, X and deltaX also need to be defined.

➢ We will add X and deltaX in the description of variables.

**Results and Discussion**

**Figures**

Fig. 3: To improve readability, I suggest to name the parameters with their extended names and the short name in parenthesis, e.g. gross primary production (GPP), directly in the plot and not in the caption.

➢ We thank the reviewer for the suggestion. We tried both versions of the figure but believe the abbreviations used are common enough to be understood without having to use the full name which would make the figure too crowded.

Fig. 5: It is not clear whether the x-axis ticks refer to the beginning/midday/end of the months. Moreover, more ticks would help reading the timing of events, e.g. when is full canopy development reached.

➢ We will insert additional ticks and improve the clarity of the figure in such way:

Fig. 6: According to Zanotelli et al., 2019 (section 2.1), yields in 2015 has been 63 t ha-1. Please, double check.

➢ We will double check this number to clarify if there has been a mistake.

**Line-specific comments**

L368-370: why "primarily". Isn't it all allocated to those organs? In the methods it is stated that storage Carbon is used for growth of all organs except fruits in the first 50 days after bud breaks. Moreover, from Fig.4 it looks like growth is supported by storages way beyond early May, rather until early June. When the fruit curve is already taking off.

➢ Yes correct, we will remove "primarily" as it is indeed misleading. Storage growth continues until early May only (it is not equivalent to the reaching of maximum LAI). The ticks refer to the start of the month, which may be the reason for the confusion. We will adapt the figure for more clarity.

L372: In Fig. 4, leaf biomass seems to reach the plateau earlier, in June. The peak in July better refers to observations, correct?

➢ Thanks for pointing this out, we will replace this by "mid June" and clarify that this refers to the simulated values.

L390: for clarity, replace "light pruning" with "a lighter pruning compared to the previous year" or similar. Moreover, if such lighter pruning happens on-field every second year, it should not sound like it was an extraordinary event in 2011 that cannot be captured by the model, but rather a flexibility in management that is not well represented in the model. If the model with fixed management "sees" an alternation of "good" and "bad" years, it could mean that it represents processes well, and it has a too simplified management that leaves room for improvement.

➢ We'll replace as suggested. To respond to the reviewer comment: The described practice is performed according to the farmer's assessment and does not follow a regular frequency (i.e. every second year). As such, the pruning remains a dynamic and somewhat subjective assessment of the farmer and information on the amount of pruning is usually not available. In addition, other apple varieties or types of deciduous fruit trees do not exhibit such behaviour which was another reason to consider the pruning amount as a fixed proportion of seasonal stem growth. However, further development may be considered in the future as the model is tested and applied more extensively.

L407-409: Not clear. Usually management should aim at reducing yield variability for both arable and perennial crops, e.g. irrigation to reduce precipitation variability, pruning to reduce alternate bearing of fruit trees, etc.

➢ We thank the reviewer for this comment. Indeed, the aim of field management is usually to reduce yield variability while poor management such as insufficient pruning or fruit thinning can result in undesired yield variability. More importantly however, yield variability is caused by the complex interaction of environmental conditions and tree physiological processes as well as small-scale heterogeneities in soil and trees. We will correct this in the manuscript.

L438: what is indicated in parenthesis? Standard deviation, range, ...

➢ Numbers in brackets represent net ecosystem exchange (NEE) as stated at the start of the sentence in L438 "Observed yearly sums of GPP (NEE) were 1.60 (-0.49),[…]".

L457-463: This paragraph is unclear and hard to follow. Please, report measured values along with observed values and vice versa. E.g. in L457, how much is Rs and its share in Reco for the simulations? Please, move "In contrast, simulated Reco for the same year [...]" of L459 right after "[...] measurements within the orchard (total soil respiration)." in L458.

➢ We will restructure the paragraph as follows: "Zanotelli et al. (2013) measured a total $R_s$ of 801±95 gC m$_{-2}$ in 2010 contributing around 90 % to $R_{eco}$, based on soil chamber measurements within the orchard (total soil respiration). The comparison with parallel measurements in a trenched plot produced a high ratio $R_h/R_s$ of 0.77 for the apple orchard. In contrast, simulated $R_s$ was 510 gC m$_{-2}$ contributing merely 45 % to $R_{eco}$ for the same year with a ratio $R_h/R_s$ of 0.87. Simulated $R_{eco}$ was instead dominated by autotrophic respiration ($R_a$) due to high C costs for maintenance, mainly of leaf biomass (data not shown). Other studies found that $R_s$ contributed 56-67 % to $R_{eco}$ in irrigated citrus orchards of different ages (Martin-Gorriz et al., 2020) and >60 % in forest ecosystems where the magnitude of ecosystem fluxes is generally comparable to orchards (Lasslop et al., 2012; Zanotelli et al., 2013)."

L472: The representation of the different components of respiration in CLM should be explained in the methods, as this is one of the metrics to evaluate the new model implementation.

➢ We will include a short explanation of respiration components in CLM5 in section 2.1 of the methodology.

L462 & L478: It is not clear why citrus orchards should be a valid reference also for apple orchards. The discussion needs to be improved, bringing more references (e.g. on more tree species) if existing or justifying why citrus trees can be a good reference

➢ We thank the reviewer for the comment. We use citrus orchards along with other orchards such as olive (L465) as well as natural vegetation (L462) for comparison to the studied apple orchard for the lack of existing studies of respiration components in apple orchards while citrus and olive orchards are somewhat better studied. Generally, different types of orchards share common management practices such as use of heavy machinery, irrigation, fertilization, tree pruning, and mulching that have a strong influence on soil respiration components. Furthermore, structural similarities (planting in tree rows) and the fate of carbon (e.g. storage in woody organs, allocation to fruit) are other common features of different types of orchards. As such we believe it is reasonable to use them for comparison especially since we refer to relative contribution of $R_s$ to $R_{eco}$ and not to absolute values of respiration that may indeed show more pronounced differences between species. To strengthen the discussion, we will include some of the above made arguments and other tree species (if further literature is available) in the manuscript.

L536: In the figure soil moisture (SM) is called soil water content (SWC). Please, be consisent.

➢ We will use soil moisture (SM) throughout the manuscript and thus adapt the figure accordingly.

---

## Author Comment (AC2)

**Answer to Reviewer #2 of manuscript for GMD:**

"CLM-FruitTree: A new sub-model for deciduous fruit trees in the Community Land Model (CLM5)" (Dombrowski et al.)

➢ We thank the reviewer for taking the time to review our manuscript and for providing valuable feedback on our work. Below we provide the preliminary responses to the comments. The detailed revision and resubmission of the paper will follow in a timely manner. In the meantime, we hope the reviewer will find the comments addressed appropriately.

**General comments**

In this manuscript, Dombrowski et al. introduce a new kind of crop to the Community Land Model (CLM): fruit trees. They present a parameterization for apple trees but note that the code they've written could be applied to other fruit-bearing trees. This represents an important step forward for CLM, which, like many global gridded crop models, has heretofore mostly excluded anything woody or perennial. Incorporating this development into CLM, especially with additional types of fruit trees, would enable the simulation of crops important not just for food security in terms of calories, but also in terms of nutrition and economic productivity.

➢ We appreciate that the reviewer acknowledges the value of expanding the capabilities of crop modelling in land surface models such as CLM5 to study different aspects of food security and productivity of perennial crops.

The model performs well compared to observations in terms of most evaluated metrics, especially yield. The authors do a good job in most cases of identifying discrepancies and suggesting hypotheses for their causes, which are often structural issues with CLM which it would be outside the scope of this work to resolve. The manuscript does unfortunately use just one real-life orchard for parameterization and evaluation of the model; fully incorporating apples as a scientifically-supported crop within CLM will likely take more effort to generalize the parameterization. But the work presented here represents a significant enough advance that it does merit publication in *GMD*. Importantly, the authors performed and presented the results of a basic sensitivity analysis, which will aid in future parameterization work.

➢ We are pleased that the reviewer confirms the good model performance we achieved with the CLM-FruitTree development. While this work focuses on the sub-model description, the reviewer is right in pointing out that the validity of CLM-FruitTree should be further tested in future studies by using similar datasets from other geographic regions, and possibly with longer time series and different orchard types. As pointed out by the reviewer, another challenge is represented by certain general structural issues of CLM5 that need improvement but are beyond the scope of this manuscript and should be accomplished by future studies. We will have a further look into our text to see if such challenges could be better outlined.

The manuscript is laid out logically, well-written, and well-supported by the provided figures. Most of my suggestions are relatively minor, and thus I recommend this manuscript be *published pending minor revisions*.

> ➤ In the following we present our responses to the reviewer's specific comments and technical corrections.

**Specific comments**

My only really substantive comments have to do with the exploration of discrepancies between the simulation and observations:

• L390-395: The simulated LAI in 2011 is too low, which the authors suggest could be due to pruning having been performed in the real world. But is the "alternate bearing behavior" something the authors actually expected the model to represent? If so, how? It seems like something that would need to be explicitly coded in.

> ➤ In the manuscript, we argue that the underestimation of LAI in 2011 is mainly due to a smaller C transfer from storage and lower solar radiation early in the growing season which led to lower simulated LAI. The discrepancy between observed and simulated LAI in this year may be exacerbated by a light pruning of the trees in the previous winter (compared to the normal amount of pruning) leading to a higher leaf biomass and higher observed LAI. This practice can sometimes be adopted by the farmer in an effort to manage the alternate bearing of the Fuji variety. However, such practice is not always successful, nor does it follow a regular frequency and the pruning is based on a somewhat subjective assessment of the farmer. Since information on the amount of pruning is usually not available and since other apple varieties and types of deciduous fruit trees do not exhibit such behaviour, we chose to represent the pruning as a fixed proportion of the seasonal stem growth. However, further developments may be considered in the future as the model is tested and applied more extensively. We will improve the paragraph by better explaining the effect of pruning and alternate bearing as well as their respective representation in the development in the revised version of the manuscript.

• L405-425: I would think that real-world management practices such as fruit thinning have the aim of *reducing* interannual variability (IAV), but it sounds like the authors are suggesting that CLM's IAV is too low because they're *not* represented. In general, it seems like missing physiological processes and/or extreme event representation would be more to blame for too-low IAV.

> ➤ We thank the reviewer for this comment and agree that the argumentation in the manuscript is not conclusive at this point. Indeed, the aim of field management is usually to reduce yield variability while poor management such as insufficient pruning or fruit thinning can result in undesired yield variability. More importantly however, yield variability is caused by the complex interaction of tree physiological processes and environmental conditions (e.g., frost, drought, hail, pests) some of which are

missing or no represented well in CLM5. We will improve our manuscript in this respect and provide a better description of physiological processes and/or extreme event representation affecting the interannual variability.

• L472–480: It's unclear from the data presented here that autotrophic respiration actually is too high in CLM5. Yes, it's too high a proportion of total ecosystem respiration, but the authors have established that soil respiration is too low. This paragraph should discuss absolute units in addition to relative ones.

> Thank you for this comment. We will add and discuss absolute units of Ra in the revised manuscript to make this point clearer.

In addition, some general comments:
• Please consider making your parameterization script(s) available as well.

> Thank you for your suggestion. The parameters were adjusted one-at-a-time through a mostly manual process. Therefore, the potential of the scripts for reuse or creation of a more automated parameterization script for CLM5 is limited and we thus do not consider them to bring much added value to the published code.

• According to *GMD* rules, the title needs a version number for CLM-FruitTree. Ideally this would correspond to a release tag in the GitHub repository.

> The GMD website states: "If the model development relates to a single model then the model name and the version number must be included in the title of the paper." which is the case for our title. However, for the sake of clarity, we could change the name of the new sub-model to "CLM5-FruitTree" which corresponds to the release tag of "CLM5_FruitTree" in Github.

**Technical corrections and minor comments**

• L17: EC is undefined
  > We will define EC in the revised manuscript.

• L33: Apostrophe should be a comma
  > We will make the suggested correction in the revised manuscript.

• L57: Adding abbreviation of "(LPJmL)" might be useful
  > The abbreviation will be included in the revised manuscript.

• L67: "buildup" would be clearer than "deposition"
  > We will replace "deposition" with "accumulation" in the revised manuscript.

- L92 and throughout: Should also cite Lombardozzi et al. (2020, *JGR: Biogeosci*: "Simulating Agriculture in the Community Land Model Version 5"), in addition to/instead of Lawrence et al. (2018)
  - ➢ We will add the suggested citation in L92 and L105 of the revised manuscript.

- L116-7: "active growth in the current season" is unclear
  - ➢ We will modify this part of the sentence as follows: "separating the growth from C reserves of the previous year, and photosynthetic growth of the current season".

- L144: "full bloom" is unclear
  - ➢ We will replace "after full bloom" with "at the end of flowering" to be clearer.

- L150 (Fig. 1):
  - o "brown" would be more accessible than "ochre" for non-native English readers
    - ➢ We will make the suggested change in the revised manuscript.
  - o "DISPLAY" is unclear. Is this a standard CLM term? If so, define it; if not, another word would be better.
    - ➢ The display carbon pool is a standard CLM term. Following a similar comment of referee #1, we will add a definition of the terms "display", "storage", and "transfer" pool in section 2.1 of the revised manuscript.
  - o Unclear from this that each plant part has its own storage and transfer pool (except, presumably, fruits)
    - ➢ Thank you for the suggestion. We went through multiple iterations of this figure to represent all important processes while still keeping it readable and not too congested. Finally, we decided to sketch the individual pools only in the display pool as only here different fates (e.g. flux to litter or harvest pool) apply to them. For the other two pools, we therefore only made a remark in the figure caption stating that the same components can be found in transfer and storage pool. A possible improvement could be editing the legend of the figure as follows:

[Figure]

- L162-5: Is *all* the C in storage pools transferred over the 50 days? If not, what "portion" is?
  - ➢ A portion of 0.5 is transferred out of the storage pool over the 50 day period based on the assumption that resources are partially mobilized to support growth of the new season but lacking more specific knowledge on that fraction. This is the default fraction used by CLM5 in the seasonal deciduous trees algorithm. We will add this information to the revised manuscript.

- L169-70: Are these GDD parameters something that can be set for each fruit tree PFT, or are hard-coded?
  - ➢ Yes, the GDD parameters as well as all other parameters listed in the table of Appendix C are part of the crop/PFT parameter file and thus can be adjusted by the user. We will clarify this in the revised manuscript.

- L173: "offset"? Is this the same as senescence?
  - ➢ Yes, offset is synonymous with senescence, we will consistently use senescence throughout the revised manuscript.

- L180 (Fig. 2)
  - o What are the bars, exactly? Period of growth?
    - ➢ The bars correspond to the time any plant organ is present on the field, i.e. coarse roots and stem are the woody perennial parts of the plant that remain on the field throughout the orchard lifetime while leaves and fine roots are shed and newly grown each season, and fruits are removed by harvest. As the tree is dormant there is no growth of any of the plant organs outside the growing period. We will provide additional explanation of the coloured bars in the figure caption in the revised manuscript.
  - o Would be clearer and more consistent for "canopy development" to just be "leaves"
    - ➢ We will edit the figure as suggested in the revised manuscript.

- L211-2: "CLM-FruitTree adopts the same N retranslocation strategy as used in the BDT phenology," but above (L149) it says "minor adaptations" were made.
  - ➢ The minor adaptations in the script include only the addition of the flag for perennial crops so that the N retranslocation strategy is used, but no changes to the strategy itself were made. We will make this clearer in the revised manuscript.

- L228: "effects" should be "affects"
  - ➢ Will be corrected.

- L270: Was the forcing de-trended during spinup?
  - ➢ The dataset was not de-trended during spinup. The CRUNCEP atmospheric forcing dataset used for spin-up is specifically designed to drive the community land model over a long period. It is the combination of two existing datasets and has been used for multiple studies including vegetation growth and gross primary production with CLM.

- L393-5: "In consequence to" should be "Due to" or "As a consequence of".
  - ➢ We will replace with "As a consequence of".

- L399-400: This sentence is unclear.
  - ➢ The sentence will be changed as follows: "Another reason could be some premature leaf fall in the summer as observed during field sampling."

- L405: Delete "at".
  - ➢ Will be corrected.

- L437: "Returns **to** positive"
  - ➢ Will be corrected.

- L520: "phenomena" should be "phenomenon".
  - ➢ Will be corrected.

- L531-2: This sentence is unclear. "Patchy" what?
  - ➢ We will replace "patchy" with "heterogenous (grass-covered alleys between tree rows)".

- L579-580: But also overestimation of soil respiration!
  - ➢ Soil respiration was in fact underestimated by the model as discussed in L457-471. Simulated autotrophic respiration mainly of leaf maintenance was higher than the observed values as discussed in L471-480. Both aspects are again taken up on in the conclusion L579-584. For more clarity the sentence in L 579 can be extended to: "The model exhibited small biases in NEE and $R_{eco}$ that were most likely caused by the overestimation of $R_a$, especially leaf maintenance respiration, and an underestimation of $R_s$."

- L588-9: What about pruning and fruit thinning?
  - ➢ As addressed in the "Specific comments" section, the particularities of the Fuji variety regarding alternate bearing behaviour pose a challenge to the implemented pruning, while the implementation may be sufficient for most other apple cultivars and fruit tree species. However, future developments could be envisioned once the model is further tested and used. Fruit thinning implementation would be a greater challenge to the current model structure as apples are not represented as individual fruits but rather one fruit pool. Therefore, it may be more feasible to account for this effect through parameterization of the carbon allocation to fruits instead of explicitly implementing this process. We will amend this section of the conclusion with some remarks to these two processes.

- L630 (Fig. B1): Please use a thicker font for this (or maybe a higher-res image); it disappears at medium zoom levels.
  - ➢ We will improve the readability of Fig. B1 in the revised manuscript.

- L635 (Fig. B2): Same issue as Fig. B1.

➢ We will improve the readability of Fig. B2 in the revised manuscript.

---

## Author Response (AR1)

Dear Dr. Christoph Müller,

We appreciate the careful review and the valuable and constructive feedback from the reviewers that helped to improve the quality of our manuscript.

We have carefully addressed and incorporated the comments and suggestions made by both reviewers. Specifically, we more clearly state the assumptions and limitations of our model development. Further major changes to the manuscript included some additions and restructuring of section 2.1 and section 2.2 of the methods. Additionally, we strengthened the discussion in a few points, e.g. inter-annual yield variability, the implemented pruning practice, and orchard canopy structure. Finally, we corrected and clarified all minor inconsistencies in the text and made small improvements to some figures.

Apart from the changes based on the reviewers' comments, we now included additional reference to Fan et al. (2015) and the unpublished code of CLM-Palm, as some parts of the technical implementation of CLM5-FruitTree were based on this work which was not clearly mentioned in the old version of our manuscript.

Lastly, some minor updates to the source code were made, including specific references to Fan et al. (2015 and unpublished code) and the addition of two formerly hard-coded parameters to the parameter file.

Please find below the detailed responses to the reviewers' comments including citations from the revised manuscript.

We once again thank you and the reviewers for the time and effort invested in our manuscript. We believe our work was further improved and we thank you in advance for your consideration of the revised manuscript to *Geoscientific Model Development*.

Sincerely,

Olga Dombrowski on behalf of all co-authors

**Answers to the comments by reviewer #1**

This manuscript describes the development of a fruit-tree sub-model as part of CLM5, a well-established land and vegetation model. As pointed out by the authors, the inclusion of new agricultural vegetation types in large scale simulation models is an important advancement for understanding and quantifying their role in many biophysical earth-system processes as well as improve the representation of the agricultural sector production. Overall, the manuscript is of good scientific quality.

One limitation of the study is that it is performed on a single point for only a few years, which limits the possibility to evaluate its validity under different conditions. On the other hand, an extremely rich dataset of measurements is used to calibrate and validate the new model. This gives confidence on the representation of processes, such as GPP, NPP, Carbon allocation and crop yields.

The results are well presented with a good structure and informative figures. Yet, some aspects have not been covered, hampering full understanding of the conceptual model and the reproducibility of results:

Thank you very much for your constructive comments and suggestions. Please find our detailed responses in the following.

One of the greatest challenges in modelling orchards is the representation of the canopy structure, which is not closed and uniform in space, as usually assumed for arable crops and for natural forest. This is a crucial aspect that affects the way radiation is intercepted by the crop canopy. In the paper, it is not described what assumptions have been made regarding radiation interception and whether changes to the CLM5 model have been necessary.

Thank you for this comment. Canopy structure is indeed a crucial aspect of modelling radiation interception and energy partitioning within the crop canopy. We have therefore added further information about the assumptions made for our development in the methods description in sections 2.2 and 3.2.2:

L151-156: "The sub-model development does not include any changes to the existing calculation schemes for radiative transfer or momentum, heat, and water fluxes to explicitly account for the discontinuous canopy structure of tree rows and vegetated or non-vegetated alleys in fruit orchards. Inrow and between-row planting distances and alley vegetation are not defined directly. Instead, the orchard structure and the area covered by the canopy is accounted for through parameterization of the leaf and stem area indices, the planting density, maximum canopy height, and aerodynamic parameters, similar to the implementation of crops and forest in CLM5."

L639-644: "Currently CLM5 is still limited to the assumption of a closed canopy structure that is uniform in space hence biases in the simulated energy balance components are likely to arise from this model limitation. Future developments towards integrating multi-layer schemes for canopy processes and the explicit representation of the canopy to improve the related processes are desirable for a more realistic representation of the orchard canopy structure."

It is not mentioned what are the structural characteristics accounted for to represent the orchard in the model. Particularly:

**The planting density:**

Thank you for this comment. The planting density is now mentioned as part of the orchard structure in the revised manuscript. We also changed planting density from a hard-coded value in the source code to a user-defined parameter (*nstem*) in the parameter file, its specific value can be found in Table C1.

L154-155: "[...] the orchard structure and the area covered by the canopy is accounted for through parameterization of the leaf and stem area indices, the planting density, [...]"

L364-366: "Structural and morphological parameters such as maximum tree height (*ztopmx*), planting density (*nstem*), the ratio of stem height to radius at breast height (*taper*), or rooting depth (*root\_dmx*) were adjusted based on site-specific information (Zanotelli et al., 2013)."

**The in-row and between-rows planting distances and the ground covered by the canopy:**

We now discuss the definition of ground cover in the revised version of the manuscript:

L153-156: "In-row and between-row planting distances and alley vegetation are not defined directly. Instead, the orchard structure and the area covered by the canopy is accounted for through parameterization of the leaf and stem area indices, the planting density, maximum canopy height, and aerodynamic parameters similarly to crops and forest in CLM5."

How large are seedlings transplanted from the nursery and what is their allometry at establishment (e.g. tree height, stem diameter, LAI, sapwood/heartwood partition, ...)?

We added more detail regarding the transplanting of seedlings in sections 2.2.2 and 2.3.3:

L234-239: "A user-defined initial biomass can be assigned to leaf and fine root transfer pools via the *transplant* parameter (Table C1), while additionally 10 % of this biomass is assigned to the dead stem pool to define an initial stem area index > 0. Each pool is also assigned the corresponding amount of N. Adjustments to this parameter have only little effect on the biomass growth and yield of the adult trees as the trees reach their maximum canopy height and develop their full LAI within the first couple of years after transplanting."

L367-368: "Initial biomass at transplanting was assumed 5 gC m-2 resulting in an initial tree height of around 100 cm and a stem diameter of 16 mm. As seedlings are dormant at the time of transplanting, their LAI is 0."

An important process of fruit-tree species is flowering, which is not explicitly represented in the CLM-FruitTree model. Although, it is clearly an acceptable simplification in this kind of model, the assumptions behind this choice (e.g. optimal pollination, compensation effects between fruit numbers and size, ...) and its implication should be presented and discussed.

Thank you for this comment, we now provide a more detailed justification of the simplifications in the revised manuscript:

L179-186: "Fruit trees usually start flowering 3–4 weeks after bud break, which is not specifically represented by CLM5-FruitTree which instead assumes that fruit growth begins at the end of flowering (Lakso et al., 1999). The implementation of flowering to include effects of non-optimal pollination, frost during flowering, or hormonal processes affecting fruit set and development is outside of the scope of this development and of minor importance for large scale simulations and processes at ecosystem level that are typically the focus of land surface models such as CLM5. Consequently, CLM5-FruitTree does not produce information on fruit size or number but only on total yield which we consider adequate for most applications of the model development."

**Methods**

**Structure**

In general section 2.1 gives the motivation for developing the CLM-FruitTree, but I think it would be better structured in this form: "to simulate fruit trees we need a model that does XYZ; CLM5 with its improvement is a good base for this, indeed it includes ABC; yet, it still misses ZYX that we implement in this paper." Otherwise it is not clear why you describe those aspects of CLM5.

We restructured section 2.1 based on this suggestion. However, we decided to keep a brief introduction of CLM5 and added information about its main vegetation modelling features to offer a simple overview to readers that are less familiar with the model. We then state what is needed for the simulation of fruit

trees and how the existing characterizations of broadleaf deciduous tree and annual crops can be partially used to realize the development of the sub-model. After describing the main features of the existing characterizations, we list the elements that are missing to implement fruit trees in CLM5.

Please refer to L93-141 in the revised manuscript for details.

Section 2.2 is not very informative in terms of model conceptualization. Please, use this section to (1) give an overall description of what system your model describes (e.g. what kind of apple orchard, extensive / intensive), (2) explain which components of the system should the model represent well (e.g. it should be at least good at simulating average yields and carbon stocks), (3) describe the model concept, preferably referring to the diagram displayed in Fig. 1. Here it would be a good place also to define the three C pools that are mentioned also in 2.2.2 without a proper explanation.

Thank you for your suggestion. We now define the three plant C pools in section 2.1 and improved the information content of section 2.2 in terms of model concepts:

L93-97: "Many of the C and N cycle components of CLM5 were originally derived from the Biome BioGeochemical Cycles (Biome-BGC) model (Thornton et al., 2002). Here, vegetation is represented conceptually by three different plant C and N pools that are maintained separately for the individual plant organs (leaf, live/dead stem, fine root, live/dead coarse root, and grain). The storage pools represent C and N reserves, the transfer pools serve as intermediate pools to separate fluxes in and out of the storage pools, and the display pools represent the actual growth of a given organ (Fig. 1)."

L143-166: "To resolve the model limitations discussed in Sect. 2.1, we developed a new sub-model CLM5-FruitTree to model the ecosystem processes and exchanges of energy and matter of deciduous fruit trees grown in commercial orchards with a focus on the simulation of biomass growth and yield. More specifically, for the implementation of CLM5-FruitTree, we introduced a new phenology subroutine that describes the main phenological development of fruit trees and includes triggers for seasonal orchard management practices typical under organic or conventional production. In addition, the CN allocation module as well as corresponding modules (C and N state and flux updates) were modified to reproduce the growth dynamics of fruit trees and to model the fates of C and N in the orchard system. The sub-model development does not include any changes to the existing calculation schemes for radiative transfer or momentum, heat, and water fluxes to explicitly account for the discontinuous canopy structure of tree rows and vegetated or non-vegetated alleys in fruit orchards. In-row and between-row planting distances and alley vegetation are not defined directly. Instead, the orchard structure and the area covered by the canopy is accounted for through parameterization of the leaf and stem area indices, the planting density, maximum canopy height, and aerodynamic parameters similarly to crops and forest in CLM5.

CLM5-FruitTree combines characteristics of both BDT and annual crops to simulate a perennial woody crop with a harvestable organ making use of the existing concepts of storage, transfer, and display vegetation pools described in section 2.1 (Fig. 1). Similar to the existing BDT phenology algorithm in CLM5, the fruit tree algorithm uses a perennial deciduous phenology with standing woody biomass and annual leaf shedding. During the active growth period however, the phenology and CN allocation of vegetative and harvestable organs are described by distinct growth phases and are driven by a GDD summation similar to the crop phenology. [...]"

Section 2.2.1: It is good to start off with phenology. Please, stick to that and do not mix phenology with growth processes. E.g. why is initial biomass mentioned in L157? Similar for L163. Maybe, put these into a paragraph at the end of 2.2.1, describing growth processes triggered by phenological events.

Following the comment, we rephrased and moved the respective sentences to section 2.2.2. Furthermore, we now mention specific values for the parameters *transplant* and *ndays\_stor* in section 2.3.3.

L234-237: "A user-defined initial biomass can be assigned to leaf and fine root transfer pools via the *transplant* parameter (Table C1), while additionally 10 % of this biomass is assigned to the dead stem pool to define an initial stem area index > 0. Each pool is also assigned the corresponding amount of N. Adjustments to this parameter [...]".

L243-246: "At bud break, a fraction of the C in the storage pool of all plant components, except fruits, is transferred to the actively growing C pools over a period that can be specified by the newly added parameter *ndays\_stor*. This is based on the assumption that resources are partially mobilized to support growth of new tissue (Oliveira and Priestley, 1988; Loescher et al., 1990)."

Section 2.2.3: Please, restructure the paragraph L219-232 to make clear what is common practice in the "real world" and what is implemented in the model. First explain the common practice and then what's in the model.

We restructured the paragraph as suggested:

L283-306: "Winter pruning is a common practice in fruit orchards and may be performed throughout the winter to control the shape and size of fruit trees, as well as to manage crop load (Grechi et al., 2008). In many intensive orchard production systems, residues are mulched into the soil, possibly increasing soil C sequestration (Montanaro et al., 2010; Aguilera et al., 2015). Alternatively, residues may also be exported and treated as waste (Benyei et al. 2018) or utilized for energy production (Kazimierski et al. 2021). In CLM5-FruitTree, pruning is performed as the tree enters dormancy by removing a user-defined fraction, *prune\_fr* (Table C1), of the dead stem from both storage and displayed C pools. We remove C from the dead stem pool instead of the live stem pool since the former is the main wood pool in CLM5 that receives 85 % of the C allocated to total new wood. Furthermore, the implemented live wood turnover in CLM5 converts live stem to dead stem at the end of the growing season to account for differences in maintenance respiration and C:N ratios between these tissue types (Lawrence et al., 2018). Hence, the live stem C pool remains rather small and stable over the years so that applying pruning to this pool would have little effect on total tree biomass. The pruning implemented in CLM5-FruitTree affects only the tree biomass and height that is calculated based on this biomass pool, which in turn affects the calculation of turbulent fluxes of sensible and latent heat. However, this effect is small, and since turbulent fluxes are generally low in winter, the exact timing of pruning does not play a significant role in the magnitude of these fluxes. During the first three years after planting, trees are not pruned to allow some initial stem biomass to grow. The model treats pruning residues in one of two ways to account for their possible difference in fate: (1) residues are added to the wood harvest pool and exported from the field or (2) residues are added to the woody debris pool thus feeding the litter cycle."

**Line-specific comments**

L97-99: As the names of these pools appear here for the first time, the sentence is a bit confusing. I would suggest to clarify the sentence as follows, use italic for the pool names and refer to later sections for additional details: "Once a new onset growth period is initiated, C and the corresponding N fluxes occur out of a *storage* pool, which are temporarily stored into an intermediate pool (*transfer* pool) and then gradually transferred to the *display* growth pools (see section XYZ for details)."

Thanks for pointing this out. We now define the three plant C pools (storage, transfer, display) in section 2.1. Please refer to the earlier comment on this and L94-97 of the revised manuscript.

L100: Are there other stoichiometric relationships other then C:N ratios? If yes, the sentence is fine, otherwise please, remove stoichiometric relationships.

Thanks for pointing out this redundancy. We modified the sentence as suggested:

L118-120: "During the active growth period, C and corresponding N storage pools are replenished based on specified C:N ratios of each plant organ."

**L101-102: Sounds like a repetition of L93, please merge the two.**

We merged the information in the respective sentences:

L120-122: "During leaf senescence, C and N pools feed the litter or coarse woody debris pool except for live stem and live coarse roots that are mostly retained as structural woody tissue (dead stem and dead coarse roots)."

L122: Unclear whether the management options are related to phenology management (e.g. choice of cultivar?) or to other management practices somewhat connected to phenology (e.g. pruning?).

Thanks for this comment. We clarified the sentence:

L145-148: "More specifically, for the implementation of CLM5-FruitTree, we introduced a new phenology subroutine that describes the main phenological development of fruit trees and includes triggers for seasonal orchard management practices typical under organic or conventional production."

L123-124: "were modified" is too vague. As you don't have space here to go into details, I'd suggest to be brief but explanatory, e.g. "CN fluxes and allocation were modified to fit ....".

We edited the sentence to be more precise:

L148-151: "In addition, the CN allocation module as well as corresponding modules (including C and N state and flux updates) were modified to reproduce the growth dynamics of fruit trees and to model the fates of C and N in the orchard system."

L124-128: These are very technical details and not so much part of the model conceptualization. I wonder whether it would be possible to make a separate section on "technical implementation" to describe these.

Thank you for your suggestion. We restructured the paragraph and removed some unnecessary technical details. We renamed section 2.2 to "Model conceptualization and technical implementation" as it still includes some technical information that we consider important but that is too brief to justify a separate section.

Please refer to L143-197 in the revised manuscript for details.

L135-136: This seems quite long for modern orchards. What kind of orchards are you simulating? Intensive / extensive, low / high density, what are the assumptions on the rootstock?

Thanks for pointing this out. We clarified and added more detail and references to the above points:

L168-172: "Once planted, the orchard remains productive according to a user-defined lifespan which, depending on fruit type and production system, typically ranges between 10 and 30 years (Demestihas et al., 2017; Cerutti et al., 2014). The sub-model makes no specific assumptions on the rootstock, but the effect of different rootstocks in terms of tree height and rooting depth can be set by the user via the respective parameters, *ztopmx* and *root\_dmx* (Table C1)."

**L158: Apple growth or apple-tree growth?**

The sentence was corrected to "tree growth".

**L163: how large is the portion of C transferred?**

We added an explanation of the transferred fraction in section 2.2.2

L243-247: "At bud break, a fraction of the C in the storage pool of all plant components, except fruits, is transferred to the actively growing C pools over a period that can be specified by the newly added parameter *ndays\_stor*. This is based on the assumption that resources are partially mobilized to support

growth of new tissue (Oliveira and Priestley, 1988; Loescher et al., 1990). Lacking more specific knowledge of the exact fraction, the default of 0.5 used by the seasonal deciduous phenology in CLM5 is adopted for fruit trees."

L164: Please, provide a reference or justification for the 50 days assumption.

We added a justification for this in section 2.3.3:

L357-359: "The length of the period where growth is supported out of reserves (*ndays\_stor*) was calibrated based on the biomass measurements and the estimate by Zanotelli et al. (2013) that apple trees use stored carbohydrates in the first two months after bud break."

L165-167: From this description ("fruit starts 4-5 w after bud break", "leaf senescence occurs after harvest") it does not seem that leaves and fruits development are independent from each other.

Thanks for pointing this out. We improved the description in the revised manuscript:

L213-216: "Outside the dormant period, leaf and fruit development occur in parallel but with a time shift as fruit growth typically starts 4–5 weeks after bud break while canopy development continues until mid-season and leaf senescence does not occur until after the fruits are harvested (Wünsche and Lakso, 2000; Goldschmidt and Lakso, 2005) (Fig. 2)."

**L186: Shouldn't "except for fruits where all allocated C is assigned to the displayed pool" be part of the previous sentence?**

The sentence was changed as suggested:

L240-243: "Throughout the growing period until harvest, 5 % of the newly assimilated C is allocated to the storage pools, as defined by the *fcur* parameter, except for fruits where all allocated C is assigned to the displayed pool. For all other organs, the remaining C is also allocated to the displayed C pools."

L199-200: Allocation to fine roots and stem decline, not the root and stem pool themselves, right?

Thanks for pointing out this mistake. The sentence was changed to "Allocation to fine roots and stem continues to decline [...]."

L210: Please, expand a bit on the N retranslocation strategy, not just by referring to Lawrence et al., 2018. Doesn't this belong to 2.2.2 as it refers to N allocation. Then you could call section 2.2.3 simply "Representation of management practices" and include here details of all managements, including the assumed orchard design (planting densities, raw arrangement, training system).

Thanks for your suggestion. We expanded on the explanation of N retranslocation in section 2.2.2 and changed the name of section 2.2.3 as suggested to "Representation of management practices":

L267-271: "Fruit trees, similar to other deciduous species, have been observed to translocate N out of senescent leaves to be reused by other tree organs (Millard, 1996; Malaguti et al., 2001; Millard et al., 2006). Therefore, CLM5-FruitTree adopts the same N retranslocation strategy as used in the BDT phenology during which N is removed from falling litter based on leaf and litter C:N ratios and the available C to pay for the extraction of N from increasingly recalcitrant litter pools. Subsequently it is transferred to the plant N pool from where it can be used for the growth of new plant tissue (Lawrence et al., 2018)."

L220: What do you mean by "dead stem"? Usually pruning is meant to remove living branches. Might be that CLM does not explicitly distinguish stem and later branches. Yet, more explanations are needed here to justify the implemented pruning routine.

Thanks for your comment. We realize that the concept of dead stem was not clearly explained in the manuscript. We therefore included further details in the revised version:

L294-299: "We remove C from the dead stem pool instead of the live stem pool since the former is the main wood pool in CLM5 that receives 85 % of the C allocated to total new wood. Furthermore, the implemented live wood turnover in CLM5 converts live stem to dead stem at the end of the growing season to account for differences in maintenance respiration and C:N ratios between these tissue types (Lawrence et al., 2018). Hence the live stem C pool remains rather small and stable over the years so that applying pruning to this pool would have little effect on total tree biomass."

**L313: for clarity, X and deltaX also need to be defined.**

We added a description of X and deltaX:

L396-397: "where X is a simulated value of the control or a perturbation run,  $\Delta X$  is the summed absolute difference between the control and the perturbation run across all perturbations, [...]"

**Results and Discussion**

**Figures**

Fig. 3: To improve readability, I suggest to name the parameters with their extended names and the short name in parenthesis, e.g. gross primary production (GPP), directly in the plot and not in the caption.

We thank the reviewer for the suggestion. We tried both versions of the figure but believe the abbreviations used are common enough to be understood without having to use the full name, which would make the figure too crowded. Consequently, the figure remains unchanged in the revised manuscript.

**Fig. 5: It is not clear whether the x-axis ticks refer to the beginning/midday/end of the months. Moreover, more ticks would help reading the timing of events, e.g. when is full canopy development reached.**

Thanks for the suggestion. We inserted additional ticks and changed the labels of the figure. The figure caption was updated to clarify that we are showing daily values for the simulations and observed LAI data collected once a month. Since we are simulating complete years, simulations (in orange) start on January 1st thus x-axis ticks refer to the beginning of the month.

Figure 1: Simulated daily leaf area index (LAI) between 2010 and 2012 together with observations (±standard error) of LAI that were made once a month for the same period. Ticks on the x-axis refer to the beginning of the month.

**Fig. 6: According to Zanotelli et al., 2019 (section 2.1), yields in 2015 has been 63 t ha-1. Please, double check.**

Thanks for pointing this out. We double-checked this number with our co-author D. Zanotelli and found that our numbers are correct. The mistake was made when 63 was accidentally taken as yield while it is the number that resulted when calculating the C content of the fruit (6337 kg C ha-1 or 633 g Cm-2). Our co-author will contact the journal in an effort to correct the value in Zanotelli et al. 2019.

**Line-specific comments**

L368-370: why "primarily". Isn't it all allocated to those organs? In the methods it is stated that storage Carbon is used for growth of all organs except fruits in the first 50 days after bud breaks. Moreover, from Fig.4 it looks like growth is supported by storages way beyond early May, rather until early June. When the fruit curve is already taking off.

Thanks for the comment. We removed "primarily" and edited the second part of the sentence for more clarity

L454-455: "[...] growth is supported by C and N reserves until the start of fruit growth in early May (50 days according to *ndays\_stor* parameter)."

L372: In Fig. 4, leaf biomass seems to reach the plateau earlier, in June. The peak in July better refers to observations, correct?

We corrected the sentence:

L457-458: "Simulated leaf biomass peaks in mid-June and remains constant thereafter [...]".

L390: for clarity, replace "light pruning" with "a lighter pruning compared to the previous year" or similar. Moreover, if such lighter pruning happens on-field every second year, it should not sound like it was an extraordinary event in 2011 that cannot be captured by the model, but rather a flexibility in management that is not well represented in the model. If the model with fixed management "sees" an alternation of "good" and "bad" years, it could mean that it represents processes well, and it has a too simplified management that leaves room for improvement.

In response to the reviewers' comment, we edited the paragraph and provided additional explanation on the pruning practice in relation to alternate bearing:

L475-485: "The discrepancy between the low simulated LAI and the high observed LAI in 2011 could have been further exacerbated by a lighter pruning performed in the previous winter, compared to other years (Zanotelli et al., 2013). Such practice is sometimes performed in an attempt to counteract the strong alternate bearing behaviour of the Fuji variety, which causes a substantial drop in yield following a high yielding year (Belleggia et al., 2009; Atay et al., 2013; Pasa et al., 2021). As a consequence of the light pruning, a larger amount of vegetative and flower buds remained on the tree leading to more growth and, possibly contributing to the larger discrepancy between relatively high observed LAI and relatively low simulated LAI. The adjusted pruning is however based on a somewhat subjective assessment of the farmer and information about the exact amount is hardly available. Thus CLM5-FruitTree currently adopts a simplified pruning practice based on the removal of a fixed portion of the seasonal stem growth which manages tree size and total woody biomass without affecting LAI."

L407-409: Not clear. Usually management should aim at reducing yield variability for both arable and perennial crops, e.g. irrigation to reduce precipitation variability, pruning to reduce alternate bearing of fruit trees, etc.

Thanks for pointing this out. We clarified the discussion on yield variability:

L498-505: "While simulated yield varied between 61 and 76 t ha-1, the observations showed a greater inter-annual variability (IAV), as exemplified in the case of the year 2012 (low yield of 51 t ha-1) and 2015 (high yield of 101 t ha-1) (Fig. 6). Low IAV of yield has also been observed in previous crop simulations with CLM5 for winter wheat (Boas et al., 2021) suggesting that certain drivers of IAV, such as extreme environmental conditions (e.g., frost, heat, and hail) or plant pests and diseases and the resulting plant physiological responses (e.g., stress-induced leaf shedding or failure to flower) (Charrier et al., 2021), are missing or not represented with sufficient detail in CLM5."

L438: what is indicated in parenthesis? Standard deviation, range, ...

Numbers in brackets represent net ecosystem exchange (NEE) as stated at the start of the sentence in L537: "Observed yearly sums of GPP (NEE) were 1.60 (-0.49), [...]".

L457-463: This paragraph is unclear and hard to follow. Please, report measured values along with observed values and vice versa. E.g. in L457, how much is Rs and its share in Reco for the simulations? Please, move "In contrast, simulated Reco for the same year [...]" of L459 right after "[...] measurements within the orchard (total soil respiration)." in L458.

Based on the suggestions, we restructured the paragraph.

L556-560: "Zanotelli et al. (2013) measured a total  $R_s$  of  $801\pm95$  gC m-2 in 2010 contributing around 90 % to  $R_{eco}$ , based on soil chamber measurements within the orchard (total soil respiration). The comparison with parallel measurements in a trenched plot produced a high ratio  $R_h/R_s$  of 0.77 for the apple orchard. In contrast, simulated  $R_s$  was 510 gC m-2 contributing merely 45 % to  $R_{eco}$  for the same year with a ratio  $R_h/R_s$  of 0.87. Simulated  $R_{eco}$  was instead dominated by autotrophic respiration ( $R_a$ ) due to high C costs for maintenance, mainly of leaf biomass (data not shown)."

L472: The representation of the different components of respiration in CLM should be explained in the methods, as this is one of the metrics to evaluate the new model implementation.

Thanks for the suggestion. We included a short explanation of the respiration components in CLM5 in section 2.1:

L97-104: "C made available through photosynthesis is first used to support maintenance respiration of live organs based on organ N content, temperature, and a constant base rate as proposed by Atkin et al. (2015). Dead stem and dead coarse root components are assumed to consist of dead xylem cells, without metabolic function (no C cost for maintenance). The remaining C can then be allocated to the growth of new tissue considering associated growth respiration costs. Maintenance respiration together with growth respiration and C cost of N uptake from the soil comprise the autotrophic respiration component ( $R_a$ ) in CLM5. Plant material reaching the end of its lifespan feeds into different litter pools from where it progressively decomposes to soil organic matter under C losses through heterotrophic respiration ( $R_h$ )."

L462 & L478: It is not clear why citrus orchards should be a valid reference also for apple orchards. The discussion needs to be improved, bringing more references (e.g. on more tree species) if existing or justifying why citrus trees can be a good reference

We included a justification for the comparison to citrus orchards and natural vegetation in the revised manuscript and removed the second comparison to citrus orchards as it was indeed not meaningful in this case. We could not include additional references for a lack of data on apple orchards or other deciduous orchards. The need for further research and data is now addressed with an additional sentence at the end of the paragraph:

L561-566: "Other studies found that  $R_s$  contributed 56-67 % to  $R_{eco}$  in irrigated citrus orchards of different ages that share common management practices (i.e., use of heavy machinery, irrigation, fertilization, tree pruning, and mulching) as well as structural similarities (e.g. planting in tree rows) with the studied apple orchard. Both aspects have a strong influence on soil respiration components in orchards (Martin-Gorriz et al., 2020). In forest ecosystems, where the magnitude of ecosystem fluxes was found to be somewhat comparable to orchards,  $R_s$  contributed > 60 % to  $R_{eco}$  (Lasslop et al., 2012; Zanotelli et al., 2013)."

L586-588: "Further work and more experimental data are needed to better understand the differences in modelled and observed respiration partitioning and to improve the performance of CLM5-FruitTree to adequately simulate the respiration components in fruit orchards."

L536: In the figure soil moisture (SM) is called soil water content (SWC). Please, be consistent.

We use soil moisture (SM) throughout the revised manuscript and adapted the titles in the Figure 9 and its caption accordingly.

**Answers to the comments of reviewer #2**

**General comments**

In this manuscript, Dombrowski et al. introduce a new kind of crop to the Community Land Model (CLM): fruit trees. They present a parameterization for apple trees but note that the code they've written could be applied to other fruit-bearing trees. This represents an important step forward for CLM, which, like many global gridded crop models, has heretofore mostly excluded anything woody or perennial. Incorporating this development into CLM, especially with additional types of fruit trees, would enable the simulation of crops important not just for food security in terms of calories, but also in terms of nutrition and economic productivity.

The model performs well compared to observations in terms of most evaluated metrics, especially yield. The authors do a good job in most cases of identifying discrepancies and suggesting hypotheses for their causes, which are often structural issues with CLM which it would be outside the scope of this work to resolve. The manuscript does unfortunately use just one real-life orchard for parameterization and evaluation of the model; fully incorporating apples as a scientifically-supported crop within CLM will likely take more effort to generalize the parameterization. But the work presented here represents a significant enough advance that it does merit publication in *GMD*. Importantly, the authors performed and presented the results of a basic sensitivity analysis, which will aid in future parameterization work.

The manuscript is laid out logically, well-written, and well-supported by the provided figures. Most of my suggestions are relatively minor, and thus I recommend this manuscript be *published pending minor revisions*.

We appreciate the positive comments of the reviewer and the valuable feedback on our work. Below we provide the detailed responses to the comments.

**Specific comments**

My only really substantive comments have to do with the exploration of discrepancies between the simulation and observations:

L390-395: The simulated LAI in 2011 is too low, which the authors suggest could be due to pruning having been performed in the real world. But is the "alternate bearing behaviour" something the authors actually expected the model to represent? If so, how? It seems like something that would need to be explicitly coded in.

**Thank you for this comment. To clarify such aspects, we revised and improved the discussion on LAI.**

L473-485: "The simulations reached similar values in 2010 and 2012 matching the observations, while the simulated LAI in 2011 underestimated the measurements due to a smaller C transfer from storage and lower solar radiation early in the growing season. The discrepancy between the low simulated LAI and the high observed LAI in 2011 could have been further exacerbated by a lighter pruning performed in the previous winter, compared to other years (Zanotelli et al., 2013). Such practice is sometimes performed in an attempt to counteract the strong alternate bearing behaviour of the Fuji variety, which causes a substantial drop in yield following a high yielding year (Belleggia et al., 2009; Atay et al., 2013; Pasa et al., 2021). As a consequence of the light pruning, a larger amount of vegetative and flower buds remained on the tree leading to more growth and, possibly contributing to the larger discrepancy between relatively high observed LAI and relatively low simulated LAI. The adjusted pruning is however based on a somewhat subjective assessment of the farmer and information about the exact amount is hardly available. Thus CLM5-FruitTree currently adopts a simplified pruning practice based on the removal of a fixed portion of the seasonal stem growth which manages tree size and total woody biomass without affecting LAI."

L405-425: I would think that real-world management practices such as fruit thinning have the aim of *reducing* interannual variability (IAV), but it sounds like the authors are suggesting that CLM's IAV is too low because they're *not* represented. In general, it seems like missing physiological processes and/or extreme event representation would be more to blame for too-low IAV.

Thank you for pointing this out. We corrected the discussion on IAV and added some more detail for more clarity:

L498-505: "While simulated yield varied between 61 and 76 t ha-1, the observations showed a greater inter-annual variability (IAV), as exemplified in the case of the year 2012 (low yield of 51 t ha-1) and 2015 (high yield of 101 t ha-1) (Fig. 6). Low IAV of yield has also been observed in previous crop simulations with CLM5 for winter wheat (Boas et al., 2021) suggesting that certain drivers of IAV such as extreme environmental conditions (e.g., frost, heat, and hail) or plant pests and diseases and the resulting plant physiological responses (e.g., stress-induced leaf shedding or failure to flower) (Charrier et al., 2021), are missing or not represented with sufficient detail in CLM5."

L507-513: "Additionally, C reserves accumulated in the previous year (Greer et al., 2002) and crop load management play an important role in determining the final harvest (Penzel et al., 2020). The latter includes pruning or fruit thinning to ensure optimal fruit growth and to reduce the effect of alternate bearing. The low observed yield in 2012 may be a result of such behaviour. This phenomenon and the processes involved are not universal so that different fruit trees may be bearing regularly, irregularly or biannually (Hoblyn et al., 1937; Monselise and Goldschmidt, 1982). As such, alternate bearing and its treatment through pruning or fruit thinning cannot easily be generalized and are thus not currently implemented in CLM5-FruitTree, which could have further reduced simulated IAV."

L472–480: It's unclear from the data presented here that autotrophic respiration actually is too high in CLM5. Yes, it's too high a proportion of total ecosystem respiration, but the authors have established that soil respiration is too low. This paragraph should discuss absolute units in addition to relative ones.

In response to the comment, we provide absolute values on autotrophic respiration in the revised manuscript:

L576-577: "In contrast to the underestimation of  $R_s$  in the model, the simulated  $R_a$  of 693 gC m-2 was almost twice the measured value of 372±195 gC m-2."

**In addition, some general comments:**

Please consider making your parameterization script(s) available as well.

Thank you for your suggestion. The parameters were adjusted one-at-a-time through a mostly manual process. Therefore, the potential of the scripts for reuse or creation of a more automated parameterization script for CLM5 is limited and we do not consider them to bring much added value to the published code. We thus decided to not publish any additional scripts.

According to *GMD* rules, the title needs a version number for CLM-FruitTree. Ideally this would correspond to a release tag in the GitHub repository.

For the sake of clarity, we changed the name of the new sub-model to "CLM5-FruitTree" in the title and throughout the text of the revised manuscript. This corresponds to the release tag of "CLM5\_FruitTree" in Github.

**Technical corrections and minor comments**

L17: EC is undefined

We defined EC in the revised manuscript.

L33: Apostrophe should be a comma

Done.

L57: Adding abbreviation of "(LPJmL)" might be useful

Done.

L67: "buildup" would be clearer than "deposition"

We replaced "deposition" with "accumulation" in the revised manuscript.

L92 and throughout: Should also cite Lombardozzi et al. (2020, *JGR: Biogeosci*: "Simulating Agriculture in the Community Land Model Version 5"), in addition to/instead of Lawrence et al. (2018)

We added a citation of the suggested paper in section 2.1.

L116-7: "active growth in the current season" is unclear

We changed the sentence for more clarity:

L138-139: "[...] and the separation of growth from C reserves of the previous year and photosynthetic growth of the current season; [...]"

**L144: "full bloom" is unclear**

We replaced "after full bloom" with "at the end of flowering".

**L150 (Fig. 1):**

"brown" would be more accessible than "ochre" for non-native English readers

We use "brown" instead of "ochre" in the figure caption of the revised manuscript.

**"DISPLAY" is unclear. Is this a standard CLM term? If so, define it; if not, another word would be better.**

The display carbon pool is one of the three plant C pools in CLM5. We added a definition of the terms "display", "storage", and "transfer" pool in L94-97 of the revised manuscript.

**Unclear from this that each plant part has its own storage and transfer pool (except, presumably, fruits)**

Thanks for pointing this out. We went through multiple iterations of this figure to represent all important processes while still keeping it readable and not too congested. Finally, we decided to sketch the individual pools only in the display pool as only here different fates (e.g. flux to litter or harvest pool) apply to them. We slightly adapted the figure caption for more clarity:

L199-202: "Figure 1: Schematic of the main phenology and C allocation features of the broadleaf deciduous tree and annual crop representations in CLM5 as well as the new sub-model CLM5-FruitTree. C pools within the dashed boxes are a zoom into the individual plant organs that make up the displayed C pool (the same components can be found for the other main plant pools: storage and transfer pools respectively). Carbon pools and fluxes in green were reused for CLM5-FruitTree while pools and fluxes in brown were modified or newly added."

Additionally the legend was modified as follows:

**L162-5: Is all the C in storage pools transferred over the 50 days? If not, what "portion" is?**

Thank you for the question. We added more explanation in the sentence and transferred it to section 2.2.2 for better consistency in the revised manuscript:

L243-247: "At bud break, a fraction of the C in the storage pool of all plant components, except fruits, is transferred to the actively growing C pools over a period that can be specified by the newly added parameter *ndays\_stor*. This is based on the assumption that resources are partially mobilized to support growth of new tissue (Oliveira and Priestley, 1988; Loescher et al., 1990). Lacking more specific knowledge of the exact fraction, the default of 0.5 used by the seasonal deciduous phenology in CLM5 is adopted for fruit trees."

**L169-70: Are these GDD parameters something that can be set for each fruit tree PFT, or are hard-coded?**

Yes, the GDD parameters as well as all other parameters listed in Table C1 of the Appendix are part of the PFT parameter file and can thus be adjusted by the user. For more clarity we made an addition to the sentence:

L217-218: "The thermal thresholds to reach phases (2)–(5) (Appendix C) are defined as accumulated GDDs since bud break and can be adjusted by the user via the parameter file which applies to all parameters listed in Table C1 of the Appendix."

**L173: "offset"? Is this the same as senescence?**

Yes, offset is synonymous with senescence. For consistency we changed the term "offset" to "senescence" throughout the revised manuscript.

**L180 (Fig. 2)**

What are the bars, exactly? Period of growth? Would be clearer and more consistent for "canopy development" to just be "leaves".

The bars correspond to the time any plant organ is present on the field, i.e. coarse roots and stem are the woody perennial parts of the plant that remain on the field throughout the orchard lifetime while leaves and fine roots are shed and newly grown each season, and fruits are removed by harvest. As the tree is dormant, there is no growth of any of the plant organs outside the growing period. We included a short explanation of the coloured bars in the figure caption.

We also slightly adapted the figure as suggested for more consistency:

Figure 2: Fruit tree phenological stages of (1) bud break at the end of dormancy, (2) the start of fruit growth, (3) fruit ripening, (4) canopy maturity, (5) harvest, and (6) the start of leaf senescence. The lengths of phenological stages (2)-(5) are determined by their respective growing degree-day thresholds (GDD) starting from bud break (GDDleaf=0), while stage (6) is determined by a critical temperature threshold ( $T_{crit}$ ). Coloured bars correspond to the time any plant organ is present on the field throughout a year.

L211-2: "CLM-FruitTree adopts the same N retranslocation strategy as used in the BDT phenology," but above (L149) it says "minor adaptations" were made.

Thank you for pointing this out. We clarified the sentence:

L190-192: "Other biochemical and biophysical processes such as photosynthesis, water and litter cycles, and fixation and uptake of N were not modified except for minor adaptations to the retranslocation of N and respiration to enable the use of certain parts of these scripts for the apple PFT."

L228: "effects" should be "affects"

Done.

**L270: Was the forcing de-trended during spinup?**

The dataset was not de-trended during spin-up as it is not meaningful for the spin-up of the model states in our simulations. The CRUNCEP atmospheric forcing dataset used for spin-up is specifically designed to drive the community land model over a long period. It is the combination of two existing datasets and has been used for multiple studies including vegetation growth and gross primary production with CLM.

**L393-5: "In consequence to" should be "Due to" or "As a consequence of".**

We replaced with "As a consequence of [...]".

**L399-400: This sentence is unclear.**

We clarified the sentence:

L489-490: "Another reason could be some premature leaf fall in the summer at the expense of the inner shadowed leaves, as observed during field sampling."

**L405: Delete "at".**

Done.

L437: "Returns to positive"

Done.

L520: "phenomena" should be "phenomenon".

Done.

**L531-2: This sentence is unclear. "Patchy" what?**

We replaced "patchy" with "discontinuous orchard canopy (grass-covered alleys between tree rows)".

**L579-580: But also overestimation of soil respiration!**

Soil respiration was in fact underestimated by the model as discussed in section 3.2.2. Simulated autotrophic respiration mainly of leaf maintenance was higher than the observed values. Both aspects are again taken up on in the conclusion. For more clarity, we made an addition to the sentence in the conclusions:

L690-691: "The model exhibited small biases in NEE and  $R_{eco}$  that were most likely caused by the overestimation of  $R_a$ , especially leaf maintenance respiration, and an underestimation of  $R_s$ ."

**L588-9: What about pruning and fruit thinning?**

We added some comments on pruning and fruit thinning in the conclusions.

L699-706: "While the particular alternate bearing of the Fuji variety posed a challenge in this specific study, the pruning routine that is currently implemented may be sufficient for most other apple cultivars and fruit tree species for which this behaviour is less pronounced or not exhibited. However, future developments could be envisioned once the model is further tested and applied. In addition, management practices such as mowing or soil tillage could further enhance the model capabilities of capturing the dynamics and fate of assimilated C. Fruit thinning is another common practice in orchards, but its implementation would be more challenging, as the current model structure does not represent individual fruits. This process could, however, be implicitly accounted for through parameterization of the C allocation to fruits."

**L630 (Fig. B1): Please use a thicker font for this (or maybe a higher-res image); it disappears at medium zoom levels.**

We used a larger font in Fig. B1: